# Riemannian Diffusion Adaptation for Distributed Optimization on Manifolds

**Xiuheng Wang** [1]   **Ricardo Borsoi** [1]   **Cédric Richard** [2]   **Ali H. Sayed** [3]

## Abstract

Online distributed optimization is particularly useful for solving optimization problems with streaming data collected by multiple agents over a network. When the solutions lie on a Riemannian manifold, such problems become challenging to solve, particularly when efficiency and continuous adaptation are required. This work tackles these challenges and devises a diffusion adaptation strategy for decentralized optimization over general manifolds. A theoretical analysis shows that the proposed algorithm is able to approach network agreement after sufficient iterations, which allows a non-asymptotic convergence result to be derived. We apply the algorithm to the online decentralized principal component analysis problem and Gaussian mixture model inference. Experimental results with both synthetic and real data illustrate its performance.

## 1. Introduction

In the decentralized setting, this work deals with the multi-agent optimization problem seeking *consensus* on a Riemannian manifold $\mathcal{M}$:

$$\min_{w \in \mathcal{M}} \frac{1}{K} \sum_{k=1}^{K} J_k(w), \qquad (1)$$

where $J_k : \mathcal{M} \to \mathbb{R}$ is a local risk function defined for each agent by $J_k(w) = \mathbb{E}_{\boldsymbol{x}_k}\{Q(w; \boldsymbol{x}_k)\}$ in terms of the expectation of some loss function $Q(w; \boldsymbol{x}_k)$. The computation of $J_k(w)$ is over the unknown distribution of the data $\boldsymbol{x}_k$, which makes it necessary to use a stochastic approximation for the gradient vector based on a set of independent realizations $\boldsymbol{x}_{k,t}$, observed sequentially over time. A wide range

Part of this work was done while Xiuheng Wang was a PhD student at Université Côte d'Azur. [1]Université de Lorraine, CNRS, CRAN, France [2]Université Côte d'Azur, CNRS, OCA, France [3]École Polytechnique Fédérale de Lausanne, Switzerland. Correspondence to: Xiuheng Wang <dr.xiuheng.wang@gmail.com>.

*Proceedings of the 42$^{nd}$ International Conference on Machine Learning*, Vancouver, Canada. PMLR 267, 2025. Copyright 2025 by the author(s).

of applications in machine learning, signal processing, and control can be written in the form of (1), including principal component analysis (PCA) (Cunningham & Ghahramani, 2015; Zhang et al., 2016), parameter estimation for Gaussian mixture models (GMM) (Hosseini & Sra, 2015; Collas et al., 2023), low-rank matrix completion (Boumal & Absil, 2011; Vandereycken, 2013), and deep neural networks with orthogonal constraints (Vorontsov et al., 2017).

Decentralized optimization in Euclidean spaces has been extensively studied. As such, one may consider converting the constraint $w \in \mathcal{M}$ into a cost function in the Euclidean space and solve (1) in a composite setting. Then, a distributed proximal gradient-type algorithm can be applied if the projection onto the manifold is available. However, previous studies, e.g., (Bianchi & Jakubowicz, 2012; Di Lorenzo & Scutari, 2016; Zeng & Yin, 2018), require a convex regularizer or at least the convexity of its domain. In addition, for certain manifolds, the dimension of an embedded Euclidean space can be relatively high according to the Whitney Embedding Theorem (Lee, 2013). Due to the non-convexity and non-linearity of certain $\mathcal{M}$, these decentralized algorithms may fail when dealing with problem (1).

In response to these challenges, this paper aims to introduce a general framework to solve (1) by developing fully decentralized optimization on manifolds, which directly operates on $\mathcal{M}$ by exploiting its inherent geometry. Our main contributions are as follows:

1. **Riemannian diffusion adaptation:** We devise a Riemannian diffusion adaptation strategy, which is fully intrinsic and thus can be applied to general manifolds[1]. It comprises a sequence of efficient *adaptation* and *combination* steps. In the adaptation step, a Riemannian stochastic gradient descent (R-SGD) method is used to estimate the local solution at each agent. In the combination step, the local estimates of the neighboring agents are combined on the tangent space of the manifold.

2. **Theoretical analysis:** We provide a theoretical analysis for the performance of the proposed Riemannian diffusion adaptation strategy with a constant step size.

---

[1]This strategy is not designed for a specific manifold as it does not require an embedding of $\mathcal{M}$ into a Euclidean space.

We establish that all agents will approximately converge to a network agreement (Theorem 5.12) in the sense of a decreasing geodesic distance between their estimates over the iterations. Additionally, we establish a curvature-dependent and non-asymptotic convergence result with a proper design of a Lyapunov function (Theorem 5.15).

3. **Application to various manifolds:** We tailor our algorithm to suit two instances of Riemannian manifolds, i.e., the Grassmann manifold and a product manifold involving the manifold of symmetric positive definite (SPD) matrices. We apply our algorithm to the online distributed PCA and parameter estimation of GMMs through numerical experiments on synthetic and real-world datasets. Experimental results show its performance compared to non-cooperative, consensus, and centralized solutions.

## 2. Related work

In this section, we review related works on decentralized optimization in Euclidean spaces and the recent advances in (decentralized) optimization on Riemannian manifolds.

**Decentralized optimization in Euclidean spaces:** Distributed optimization in Euclidean spaces has been extensively studied, including incremental (Blatt et al., 2007), consensus (Nedic et al., 2010) and diffusion (Chen & Sayed, 2012; Sayed et al., 2013) strategies. In particular, diffusion strategies have been demonstrated in (Sayed, 2014; Chen & Sayed, 2015) to offer improved performance and stability guarantees under constant step-size learning and adaptive scenarios. Recently, decentralized optimization has been extensively studied in non-convex environments (Bianchi & Jakubowicz, 2012; Di Lorenzo & Scutari, 2016; Lian et al., 2017; Tatarenko & Touri, 2017; Zeng & Yin, 2018; Wang et al., 2019; Vlaski & Sayed, 2021).

**Optimization on manifolds:** Riemannian optimization has garnered significant interest as it considers the geometry of manifolds, recently presented in detail in the books (Absil et al., 2009) and (Boumal, 2023). Of particular interest are stochastic optimization methods due to their efficiency and scalability. The first asymptotic convergence analysis of R-SGD was provided in (Bonnabel, 2013), highlighting diverse applications such as PCA. The first global convergence results for first-order Riemannian optimization with geodesic convexity were obtained in (Zhang & Sra, 2016). The finite-sum, stochastic setting has been further investigated in (Zhang et al., 2016; Sato et al., 2019) for variance reduction. The work (Tripuraneni et al., 2018) constructed and analyzed a variant of R-SGD that generalizes the classical Polyak-Ruppert iterate-averaging scheme. Several

stochastic Riemannian Frank-Wolfe methods were introduced in (Weber & Sra, 2022). More recently, the behavior of various stochastic optimization algorithms around saddle points in geodesically non-convex functions was studied in (Hsieh et al., 2024). In (Wang et al., 2024a), the non-asymptotic convergence of R-SGD with constant step sizes was studied and applied to the change point detection on manifolds.

**Decentralized optimization on manifolds:** The literature on decentralized optimization on Riemannian manifolds can be roughly divided into *extrinsic* and *intrinsic* methods.

The extrinsic methods are based on *induced arithmetic mean* (Sarlette & Sepulchre, 2009), and rely on the specific embedding of the manifolds in Euclidean spaces (where traditional Euclidean consensus can be employed), which is often studied for specific manifolds. For stochastic optimization, the incremental and consensus strategies have been extended to decentralized R-SGD-type on the unit sphere (Wang et al., 2023) and Stiefel manifolds (Chen et al., 2021), respectively. For the deterministic case, an augmented Lagrangian method (Wang & Liu, 2022) and a type of conjugate gradient method (Chen et al., 2024) were also designed for decentralized optimization on the Stiefel manifold. In addition, a consensus strategy has also been extended to compact submanifolds (Deng & Hu, 2023).

The intrinsic methods are based on *Fréchet mean* (Tron et al., 2012) (or center of mass) and developed with the inherent geometry of manifolds, such as geodesic distance, Riemannian gradient, and exponential mapping. These methods can be studied on more general manifolds including, but not limited to, the unit sphere, Stiefel manifolds, Grassmann manifolds, and the manifold of SPD matrices. Different distributed strategies (Tron et al., 2012; Kraisler et al., 2023a;b) were studied to achieve network agreement on manifolds. Another distributed strategy (Shah, 2017) was considered to solve (1) with a diminishing step size and two rounds of communication in each iteration. However, few methods have investigated the diffusion strategy on manifolds, though it has been proven to have superior properties in Euclidean spaces, especially in continuous learning and adaptive scenarios. A work extending the diffusion strategy to manifolds was introduced in (Wang et al., 2024b), but the algorithm is inefficient due to inner-loop optimization[2] and does not have any theoretical analyses.

Recently, another branch of distributed optimization on manifolds considering a central server was also investigated, with settings of communication efficiency (Huang & Pan, 2020) and federated learning (Li & Ma, 2023; Huang et al., 2024a;b).

---

[2]We further support this claim with a numerical evaluation in Appendix D.1

# 3. Background

This section introduces some basic concepts of Riemannian geometry, focusing on the essential tools for optimization on manifolds. Detailed presentations can be found in (Absil et al., 2009) and (Boumal, 2023).

A *Riemannian manifold* $(\mathcal{M}, g)$ is a constrained set $\mathcal{M}$ endowed with a *Riemannian metric* $g_x(\cdot, \cdot) : T_x\mathcal{M} \times T_x\mathcal{M} \to \mathbb{R}$, defined for every point $x \in \mathcal{M}$, with $T_x\mathcal{M}$ the so-called *tangent space* of $\mathcal{M}$ at $x$. A *geodesic* $\gamma_v : [0, 1] \to \mathcal{M}$ is the curve of minimal length linking two points $x, y \in \mathcal{M}$ such that $x = \gamma_v(0)$ and $y = \gamma_v(1)$, with $v \in T_x\mathcal{M}$ the velocity of $\gamma_v$ at 0 denoted by $\dot{\gamma}_v(0)$. The *geodesic distance* $d(\cdot, \cdot) : \mathcal{M} \times \mathcal{M} \to \mathbb{R}$ is defined as the length of the geodesic linking two points $x, y \in \mathcal{M}$. It satisfies all the conditions to be a metric.

The *exponential map* $w = \exp_x(v)$ is defined as the point $w \in \mathcal{M}$ located on the unique geodesic $\gamma_v(t)$ with endpoints $x = \gamma_v(0)$, $w = \gamma_v(1)$ and velocity $v = \dot{\gamma}_v(0)$. Consider a smooth function $f : \mathcal{M} \to \mathbb{R}$. The *Riemannian gradient* of $f$ at $x \in \mathcal{M}$ is defined as the unique tangent vector $\nabla f(x) \in T_x\mathcal{M}$ satisfying $\frac{d}{dt}\big|_{t=0} f(\exp_x(tv)) = \langle \nabla f(x), v \rangle_x$, for all $v \in T_x\mathcal{M}$. The *Riemannian Hessian* of $f$ at $x$ is an operator $\nabla_x^2 f$ such that $\frac{d}{dt}\big|_{t=0} \langle \nabla f(\exp_x(tv)), \nabla f(\exp_x(tv)) \rangle_x = 2\langle \nabla f(x), (\nabla_x^2 f)v \rangle_x$.

# 4. Algorithm development

Let us define the product manifold $\mathcal{M}^K \triangleq \mathcal{M} \times \cdots \times \mathcal{M}$, which is the $K$-fold Cartesian product of $\mathcal{M}$ with itself. We also define $\boldsymbol{w} \triangleq \mathrm{col}\{\boldsymbol{w}_1, \cdots, \boldsymbol{w}_K\}$ to indicate a point on $\mathcal{M}^K$. The decentralized optimization problem (1) contains an implicit consensus: the individual models are required to be common on the manifolds, i.e., $\boldsymbol{w}_k = w$, $\forall k$. One can encourage consensus on manifolds by penalizing pairwise differences between connected agents. Let us represent the $K$ agents as the nodes of a graph $\mathcal{G}$. With a natural generalization of the Euclidean case, we consider the geodesic distance-based consensus problem (Tron et al., 2012), i.e., minimization of the penalty $P(\boldsymbol{w}) \triangleq \sum_{k=1}^{K} P_k(\boldsymbol{w}_k)$ where $P_k(\boldsymbol{w}_k) \triangleq \frac{1}{2}\sum_{\ell=1}^{K} c_{\ell k} d^2(\boldsymbol{w}_k, \boldsymbol{w}_\ell)$ and $c_{\ell k} \triangleq [C]_{\ell k}$, with $C$ a weighted adjacency matrix of the graph $\mathcal{G}$, representing the strength of link between each pair of agents. For a connected graph, $P(\boldsymbol{w}) = 0$ if and only if $\{\boldsymbol{w}_k\}_{k=1}^{K}$ are equal for all $k$. This results in the following optimization problem with a constraint:

$$\min_{\boldsymbol{w} \in \mathcal{M}^K} J(\boldsymbol{w}) \qquad s.t. \quad P(\boldsymbol{w}) = 0, \qquad (2)$$

where $J(\boldsymbol{w}) \triangleq \frac{1}{K}\sum_{k=1}^{K} J_k(\boldsymbol{w}_k)$. To minimize the global cost function in (2), we follow the diffusion adaptation strategy in Euclidean spaces (Sayed et al., 2014; Yuan et al., 2018; Vlaski et al., 2023), and appeal to an incremental gradient descent argument. We first apply an R-SGD to

---

**Algorithm 1** Riemannian Diffusion Adaptation

**Input:** Step sizes $\mu, \alpha$, graph adjacency matrix $C$.
Initialize $\{\boldsymbol{w}_{k,0}\}$ for all $k$ with a random point on $\mathcal{M}$.
**for** $t = 1, 2, \cdots$ **do**
  **for** each agent $k$ **do**
    $\boldsymbol{\phi}_{k,t} = \exp_{\boldsymbol{w}_{k,t-1}}\left(-\mu\widehat{\nabla J}_k(\boldsymbol{w}_{k,t-1})\right);$
    $\boldsymbol{w}_{k,t} = \exp_{\boldsymbol{\phi}_{k,t}}\left(\alpha\sum_{\ell=1}^{K} c_{\ell k}\exp_{\boldsymbol{\phi}_{k,t}}^{-1}(\boldsymbol{\phi}_{\ell,t})\right);$
  **end for**
**end for**

---

the risk $J(\boldsymbol{w})$ and subsequently descend along the penalty $P(\boldsymbol{w})$. Using node-level quantities we have

$$\boldsymbol{\phi}_{k,t} = \exp_{\boldsymbol{w}_{k,t-1}}\left(-\mu\widehat{\nabla J}_k(\boldsymbol{w}_{k,t-1})\right), \qquad (3)$$

$$\boldsymbol{w}_{k,t} = \exp_{\boldsymbol{\phi}_{k,t}}\left(-\alpha\nabla P_k(\boldsymbol{\phi}_{k,t})\right), \qquad (4)$$

where $\mu$ and $\alpha$ are step sizes, $\widehat{\nabla J}_k$ is a stochastic approximation of the Riemannian gradient of $J_k$, and $\nabla P_k$ can be computed in an explicit form (Afsari et al., 2013), given by

$$\nabla P_k(\boldsymbol{\phi}_{k,t}) = \frac{1}{2}\sum_{\ell=1}^{K} c_{\ell k}\nabla d^2(\boldsymbol{\phi}_{k,t}, \boldsymbol{\phi}_{\ell,t})$$

$$= -\sum_{\ell=1}^{K} c_{\ell k}\exp_{\boldsymbol{\phi}_{k,t}}^{-1}(\boldsymbol{\phi}_{\ell,t}). \qquad (5)$$

The Riemannian diffusion adaptation strategy, summarized in Algorithm 1, contains two steps: an adaptation step (3) where agent $k$ uses its own data $\boldsymbol{x}_{k,t-1}$ to update its solution $\boldsymbol{\phi}_{k,t}$ and a combination step (4) where the intermediate estimates $\{\boldsymbol{\phi}_{l,t}\}$ are combined, on the tangent space of $\boldsymbol{\phi}_{k,t}$, according to the weighting coefficients $\{c_{lk}\}$ in (5) to obtain the estimate $\boldsymbol{w}_{k,t}$. Note that in the special case that $\mathcal{M}$ is a Euclidean space, we can take $\exp_x(v)$ as vector addition of $x + v$, and our algorithm reduces to the diffusion adaptation algorithm in the Euclidean space (Chen & Sayed, 2012; Sayed et al., 2013). Here we emphasize that the local update of each agent in (3) is performed by stochastic Riemannian optimization with constant step size, which plays an important role in tasks in need of continuous learning and adaptation (Sayed et al., 2013; Sayed, 2014).

# 5. Theoretical analysis

In this section, we analyze the convergence of Algorithm 1 in the constant step size setting.

In analyzing the dynamics of the distributed algorithm (3) and (4), it is useful to introduce the following stacked vector notation by collecting variables from across the network:

$$\boldsymbol{w}_t \triangleq \mathrm{col}\{\boldsymbol{w}_{1,t}, \cdots, \boldsymbol{w}_{K,t}\} \in \mathcal{M}^K$$

$$\widehat{\nabla J}(\boldsymbol{w}_t) \triangleq \mathrm{col}\{\widehat{\nabla J}_1(\boldsymbol{w}_{1,t}), \cdots, \widehat{\nabla J}_K(\boldsymbol{w}_{K,t})\} \in T_{\boldsymbol{w}_t}\mathcal{M}^K$$

$$\phi_t \triangleq \mathrm{col}\{\phi_{1,t}, \cdots, \phi_{K,t}\} \in \mathcal{M}^K$$

$$\nabla P(\phi_t) \triangleq \mathrm{col}\Big\{ -\sum_{\ell=1}^{K} c_{\ell 1} \exp^{-1}_{\phi_{1,t}}(\phi_{\ell,t}), \dots,$$
$$-\sum_{\ell=1}^{K} c_{\ell K} \exp^{-1}_{\phi_{K,t}}(\phi_{\ell,t}) \Big\} \in T_{\phi_t} \mathcal{M}^K$$

where $\mathrm{col}\{\cdot\}$ is obtained by stacking the arguments columnwise and $T_x \mathcal{M}^K$ is the tangent space of $\mathcal{M}^K$ at $x$, see Proposition 3.20 in (Boumal, 2023). We can then write (3) and (4) compactly as

$$\phi_t = \exp_{w_{t-1}}\left( -\mu \widehat{\nabla J}(w_{t-1}) \right), \qquad (6)$$
$$w_t = \exp_{\phi_t}\left( -\alpha \nabla P(\phi_t) \right). \qquad (7)$$

Step (7) can be regarded as a one-step Riemannian gradient descent with a step size $\alpha$ to approximate a global minimum of $P(\phi)$, belonging to the *consensus submanifold* $\mathcal{A}$, defined as

$$\mathcal{A} \triangleq \{\phi \in \mathcal{M}^K \mid \phi_i = \phi_j, \forall i, j\}. \qquad (8)$$

We start by introducing some technical assumptions and existing auxiliary results before presenting new results.

### 5.1. Assumptions and auxiliary results

Let us denote the *convexity submanifold* (Tron et al., 2012) of product manifolds $\mathcal{M}^K$ as $\mathcal{B} \subseteq \mathcal{M}^K$. We introduce the following standard assumptions in the literature on Riemannian optimization:

**Assumption 5.1** (**Regularization on manifold**). (Bonnabel, 2013; Zhang et al., 2016; Tripuraneni et al., 2018; Afsari, 2011) (a) The sequences $\{\phi_t\}_{t\geq 0}$ and $\{w_t\}_{t\geq 0}$ generated by the algorithm stay continuously in $\mathcal{B}$, and $J$ attains its optimum $w^*$ in $\mathcal{B}$; (b) the sectional curvature in $\mathcal{B}$ is *upper* bounded by $\kappa_{\max}$; (c) the sectional curvature in $\mathcal{B}$ is *lower* bounded by $\kappa_{\min}$; and (d) $\mathcal{B}$ is compact, and the diameter of $\mathcal{B}$ is bounded by $D$, that is, $\max_{x,y \in \mathcal{B}} d(x, y) \leq D$; (e) $D < D^*$, where $D^*$ is defined as $D^* \triangleq \min(\mathrm{inj}(\mathcal{M}), \frac{\pi}{\sqrt{\kappa_{max}}})$ with $\mathrm{inj}(\mathcal{M})$ is the injectivity radius of $\mathcal{M}$, which implies that the exponential map is invertible within $\mathcal{B}$.

Also, it is necessary to assume some properties of the weighted adjacency matrix $C$ according to which the agents interact over the graph topology $\mathcal{G}$. In addiction to the direct assumptions on $\mathcal{G}$ (e.g., left-stochastic) in Euclidean space (Chen & Sayed, 2012; Sayed et al., 2013), for distributed optimization on manifolds, we also make the following assumptions on the eigenvalues of the Riemannian Hessian of $P$, whose computation involves $C$, see Subsection 2.1.3 in (Afsari et al., 2013) and Proposition 8 in (Tron et al., 2012) for examples.

**Assumption 5.2** (**Regularization on graph**). (Chen & Sayed, 2012; Sayed et al., 2013; Afsari et al., 2013) Assume that the undirected $\mathcal{G}$ is connected and its adjacency matrix $C$ is left-stochastic, i.e., $c_{\ell k} \geq 0, \sum_{\ell=1}^{K} c_{\ell k} = 1$ for each agent $k$, denote lower and upper bounds on the eigenvalues of the Hessian of $P$ in $\mathcal{B}$ as $h_{min}$ and $h_{max}$, and suppose that $h_{min} \geq 0$.

Note this assumption implies that $P$ is geodesically (strong) convex and smooth on $\mathcal{B}$. Under Assumption 5.2, the global minimum exists and is unique if all $\{\phi_{k,t}\}_{k=1}^K$ are contained in $\mathcal{B}$, i.e., $P : \mathcal{B} \to \mathcal{A}$ is well-defined (Tron et al., 2012).

Recall the following trigonometric distance bound essential in the Riemannian optimization analysis.

**Lemma 5.3.** *(Bonnabel, 2013; Zhang & Sra, 2016) If $a, b, c$ are the side lengths of a geodesic triangle in a Riemannian manifold with sectional curvature lower bounded by $\kappa_{\min}$, and $A$ is the angle between sides $b$ and $c$ (defined through the inverse exponential map and inner product in tangent space), then*

$$a^2 \leq \frac{\sqrt{|\kappa_{\min}|}c}{\tanh(\sqrt{|\kappa_{\min}|}c)} b^2 + c^2 - 2bc\cos(A). \qquad (9)$$

We define the following key geometric constant that captures the impact of manifold curvature:

$$\zeta = \begin{cases} \frac{\sqrt{|\kappa_{\min}|}D}{\tanh(\sqrt{|\kappa_{\min}|}D)}, & \text{if } \kappa_{\min} < 0, \\ 1, & \text{if } \kappa_{\min} \geq 0, \end{cases} \qquad (10)$$

Note that most (if not all) practical manifold optimization problems can satisfy these assumptions.

Leveraging Assumption 5.1 and Lemma 5.3, we can readily establish the following corollary.

**Corollary 5.4.** *(Zhang & Sra, 2016) For any Riemannian manifold $\mathcal{M}$ where the sectional curvature is lower bounded by $\kappa_{\min}$ and for any points $x, x_t \in \mathcal{M}$, the update $x_{t+1} = \exp_{x_t}(-\mu\nabla F(x_t))$ satisfies the inequality:*

$$\langle -\nabla F(x_t), \exp^{-1}_{x_t}(x) \rangle \leq \frac{1}{2\mu}\left( d^2(x_t, x) - d^2(x_{t+1}, x) \right)$$
$$+ \frac{\zeta\mu}{2}\|\nabla F(x_t)\|^2. \qquad (11)$$

This corollary unveils a significant relationship between two consecutive updates within an iterative optimization algorithm on a manifold with curvature bounded from below.

Part of our analysis will be performed under the following assumption of geodesically convex risk functions.

**Assumption 5.5** (**Geodesical convexity**). A function $J_k : \mathcal{M} \to \mathbb{R}$ is geodesically convex (g-convex) if for any $x, y \in$

$\mathcal{M}$, a geodesic $\gamma$ such that $\gamma(0) = x$ and $\gamma(1) = y$, and $\alpha \in [0, 1]$, we have:

$$J_k(\gamma(\alpha)) \leq (1 - \alpha)J_k(x) + \alpha J_k(y), \quad (12)$$

or equivalently, we have

$$J_k(y) \geq J_k(x) + \langle \nabla J_k(x), \exp_x^{-1}(y) \rangle. \quad (13)$$

Meanwhile, we require the risk function $J_k$ at each agent to be geodesically smooth.

**Assumption 5.6 (Geodesic smoothness).** A differentiable function $J_k$ is geodesically $L$-smooth ($L$-g-smooth) if its gradient is $L$-Lipschitz, i.e., for any $x, y \in \mathcal{M}$, it satisfies:

$$J_k(y) \leq J_k(x) + \langle \nabla J_k(x), \exp_x^{-1}(y) \rangle + \frac{L}{2} \| \exp_x^{-1}(y) \|^2, \quad (14)$$

where the gradient of a function $J_k : \mathcal{M} \to \mathbb{R}$ is said to be $L$-Lipschitz if, for any $x, y \in \mathcal{M}$ in the domain of $J_k$, it satisfies:

$$\left\| \nabla J_k(x) - \Gamma_y^x \nabla J_k(y) \right\| \leq L \left\| \exp_x^{-1}(y) \right\|, \quad (15)$$

where $\Gamma_y^x$ denotes the parallel transport operator from $y$ to $x$.

In addition, we make assumptions about the average and second moment of the gradient noise process.

**Assumption 5.7 (Gradient noise process).** Denote $\mathcal{F}_t$ as the filtration generated by the random process $\boldsymbol{w}_{k,s}$ for all $k$ and for $s \leq t$, that is,

$$\mathcal{F}_t \triangleq \{\boldsymbol{w}_0, \boldsymbol{w}_1, \cdots, \boldsymbol{w}_t\}, \quad (16)$$

where $\boldsymbol{w}_s \triangleq \text{col}\{\boldsymbol{w}_{1,s}, \cdots, \boldsymbol{w}_{K,s}\}$ contains the iterates across the network at time $s$. Define $S_{t+1}(\boldsymbol{w}_t) \triangleq \widehat{\nabla J}(\boldsymbol{w}_t) - \nabla J(\boldsymbol{w}_t)$ as the gradient noise process at the time instant $t$. It is assumed that

$$\mathbb{E}\{S_{t+1}(\boldsymbol{w}_t)|\mathcal{F}_t\} = \mathbf{0}, \quad (17)$$

$$\mathbb{E}\{\|S_{t+1}(\boldsymbol{w}_t)\|^2|\mathcal{F}_t\} \leq \sigma^2, \quad (18)$$

for some non-negative constant $\sigma$.

With these assumptions, we can build some preliminary lemmas that will be used in the proof of our main results.

### 5.2. Preliminary lemmas

We first establish a lemma that bounds the gradient of the penalty function $P$ in terms of the penalty itself.

**Lemma 5.8.** *Under Assumption 5.2, for the gradient of the penalty, it holds that*

$$\|\nabla P(\boldsymbol{\phi}_t)\|^2 \leq 2P(\boldsymbol{\phi}_t). \quad (19)$$

*Proof.* Appendix A.1. □

Under Assumption 5.5, we can establish the following property for the risk function $J$.

**Lemma 5.9.** *Under Assumption 5.5, define $\bar{\boldsymbol{w}} = \text{col}\{\boldsymbol{w}_m, \cdots, \boldsymbol{w}_m\}$ with $\boldsymbol{w}_m$ being the Fréchet mean (barycenter) of $\boldsymbol{w}_1, \ldots, \boldsymbol{w}_K$, we have*

$$J(\bar{\boldsymbol{w}}) \leq J(\boldsymbol{w}). \quad (20)$$

*Proof.* Appendix A.2. □

This proof follows similarly to Proposition 10 in (Yokota, 2016) and Theorem 1.1 in (Paris, 2020).

The following lemma builds on assumptions 5.1 and 5.6, and establishes an upper bound on $\|\nabla J(\boldsymbol{w}_t)\|$.

**Lemma 5.10.** *Under assumptions 5.1 and 5.6, we have:*

$$\|\nabla J(\boldsymbol{w}_t)\| \leq G, \quad (21)$$

*for a non-negative constant $G < \infty$.*

*Proof.* Appendix A.3. □

This upper bound is similar to the one used in (Shah, 2017; Deng & Hu, 2023) under a diminishing step size.

### 5.3. Network agreement

To begin with, we first show that the Riemannian diffusion adaptation algorithm approximately converges toward network agreement. In other words, $\boldsymbol{w}_t$ converges to the consensus submanifold $\mathcal{A}$ with high probability. The following lemma builds on Lemma 5.10 under the additional conditions set forth in assumption 5.7 and 5.2.

**Lemma 5.11.** *Under assumptions 5.1, 5.2, 5.6 and 5.7, suppose $\alpha \in (0, h_{max}^{-1}]$. The sequence $\{P(\boldsymbol{\phi}_t)\}_{t \geq 0}$ satisfies the following relation:*

$$\mathbb{E}\{P(\boldsymbol{\phi}_{t+1}) - P(\boldsymbol{\phi}_t)\} \leq -\frac{\alpha}{4}\mathbb{E}\|\nabla P(\boldsymbol{\phi}_t)\|^2 + \frac{5\mu^2}{\alpha}G^2$$
$$+ \frac{\mu^2}{\alpha}\sigma^2. \quad (22)$$

*Proof.* Appendix A.4. □

This lemma reveals the evolution of the difference $\mathbb{E}\{P(\boldsymbol{\phi}_{t+1}) - P(\boldsymbol{\phi}_t)\}$ in the optimization process. The first term on the right-hand side of (22) is strictly negative and suggests a decrease in the expectation of penalty by a magnitude proportional to $\mathbb{E}\|\nabla P(\boldsymbol{\phi}_t)\|^2$. However, the second and third terms on the right-hand side of (22) could be large enough to allow the objective value to increase. In

the following, with an additional assumption that the cost $J$ is geodesically convex (Assumption 5.5), we prove that the expectation of penalty decreases strictly and can be upper bounded with a small value after sufficient iterations.

**Theorem 5.12.** *Under assumptions 5.1, 5.2, 5.5, 5.6, and 5.7, suppose $\alpha \in (0, h_{max}^{-1}]$. The sequence $\{P(\boldsymbol{\phi}_t)\}_{t \geq 0}$ satisfies the following relation:*

$$\mathbb{E}\{P(\boldsymbol{\phi}_t)\} \leq \frac{11\mu^2}{2\alpha\tau}G^2 + \frac{3\mu^2}{\alpha\tau}\sigma^2, \qquad (23)$$

*after sufficient iterations $s_o$, given by*

$$s_o = \frac{2\log(\mu)}{\log(1-\tau)} + O(1) = O(\mu^{-1}), \qquad (24)$$

*where $\tau = \min\{\frac{1}{2\zeta}, \alpha h_{min}\}$, $O(1)$ denotes a constant term, and $O(\mu^{-1})$ denotes a term that is equal or higher in order than $\mu^{-1}$, the last equality holds for sufficiently small $\mu$.*

*Proof.* Appendix B.1. □

The result in Theorem 5.12 establishes that after sufficient iterations $s_o = O(\mu^{-1})$, we have:

$$\mathbb{E}\{P(\boldsymbol{\phi}_t)\} \leq O(\mu^2), \qquad (25)$$

or, from Markov's inequality:

$$\Pr\{P(\boldsymbol{\phi}_t) \geq \mu\} \leq O(\mu), \qquad (26)$$

which means the local estimates in $\boldsymbol{\phi}_t$ coalesce around $\boldsymbol{\phi}_t^* \in \mathcal{A}$ (where $P(\boldsymbol{\phi}_t^*) = 0$) with high probability. These results are consistent with Theorem 1 in (Vlaski & Sayed, 2021) where the Euclidean diffusion adaptation algorithm is analyzed for non-convex environments.

Combined with Lemma 5.8, Theorem 5.12 leads to the following corollary.

**Corollary 5.13.** *With the assumptions in Theorem 5.12. The sequence $\{\|\nabla P(\boldsymbol{\phi}_t)\|^2\}_{t \geq 0}$ satisfies the following relation:*

$$\mathbb{E}\|\nabla P(\boldsymbol{\phi}_t)\|^2 \leq \frac{11\mu^2}{\alpha\tau}G^2 + \frac{6\mu^2}{\alpha\tau}\sigma^2, \qquad (27)$$

*after sufficient iterations $s_o = O(\mu^{-1})$.*

Hence, according to $\nabla P(\boldsymbol{\phi}_t^*) = \boldsymbol{0}$ and the update in (7), we conclude that $\boldsymbol{w}_t$ approximately approaches $\boldsymbol{\phi}_t$ and achieves network agreement, or equivalently $\boldsymbol{w}_t \in \mathcal{A}$ with high probability after sufficient iterations.

### 5.4. Non-asymptotic convergence

Next, we examine the convergence of Algorithm 1 after sufficient iterations $s_o$. For this purpose, we make use of the upper bound on $\mathbb{E}\|\nabla P(\boldsymbol{\phi}_t)\|^2$ given in Corollary 5.13. Before this, we introduce the following lemma that builds on the same assumptions as in Lemma 5.11 and can be regarded as a symmetric result of the relation (22).

**Lemma 5.14.** *Under assumptions 5.1, 5.2, 5.6 and 5.7, suppose $\mu \in (0, L^{-1}]$. The sequence $\{J(\boldsymbol{w}_t)\}_{t \geq 0}$ satisfies the following relation:*

$$\mathbb{E}\{J(\boldsymbol{w}_{t+1}) - J(\boldsymbol{w}_t)\} \leq -\frac{\mu}{4}\mathbb{E}\|\widehat{\nabla J}(\boldsymbol{w}_t)\|^2$$
$$+ \frac{5\alpha^2}{\mu}\mathbb{E}\|\nabla P(\boldsymbol{\phi}_{t+1})\|^2. \quad (28)$$

*Proof.* Appendix A.5. □

This lemma shows the evolution of the term $\mathbb{E}\{J(\boldsymbol{w}_{t+1}) - J(\boldsymbol{w}_t)\}$ in the optimization process. The strictly negative term on the right-hand side of (28) suggests a decrease in the expectation of the risk function by a magnitude proportional to $\mathbb{E}\|\widehat{\nabla J}(\boldsymbol{w}_t)\|^2$, while the positive one could be large enough to allow the objective value to increase.

In the following, we consider (and study the convergence of) a streaming average of iterates $\{\boldsymbol{w}_{s_o+1}, \cdots, \boldsymbol{w}_t\}$, given by $\{\boldsymbol{w}'_{s_o+1}, \cdots, \boldsymbol{w}'_t\}$ with $\boldsymbol{w}'_{s_o+1} = \boldsymbol{w}_{s_o+1}$, $\boldsymbol{w}'_{s+1} = \exp_{\boldsymbol{w}'_s}\left(\frac{1}{s-s_o+1}\exp_{\boldsymbol{w}'_s}^{-1}(\boldsymbol{w}_{s+1})\right)$ for $s_o + 1 \leq s \leq t - 2$, and

$$\boldsymbol{w}'_t = \exp_{\boldsymbol{w}'_{t-1}}\left(\frac{2\zeta}{2\zeta + t - s_o - 1}\exp_{\boldsymbol{w}'_{t-1}}^{-1}(\boldsymbol{w}_t)\right). \quad (29)$$

This provides a natural way of averaging along a trajectory restricted to a manifold (Tripuraneni et al., 2018). For example, when $\mathcal{M}$ is a Euclidean space, we can write $\exp_x(v)$ as $x + v$, and the streaming average reduces to $\boldsymbol{w}'_{s_o+1} = \boldsymbol{w}_{s_o+1}$, $\boldsymbol{w}'_{s+1} = \boldsymbol{w}'_s + \frac{1}{s-s_o+1}(\boldsymbol{w}'_s - \boldsymbol{w}_{s+1})$ for $s_o + 1 \leq s \leq t - 2$ and $\boldsymbol{w}'_t = \boldsymbol{w}'_{t-1} + \frac{2\zeta}{2\zeta+t-s_o-1}(\boldsymbol{w}'_{t-1} - \boldsymbol{w}_t)$. Inspired by (Zhang & Sra, 2016), we design a Lyapunov function of $\boldsymbol{w}_t$ as

$$\Delta'_t \triangleq J(\boldsymbol{w}'_t) - J(\boldsymbol{w}^*), \qquad (30)$$

with auxiliary variables $\boldsymbol{w}'_t$ defined in (29) and $\boldsymbol{w}^*$ denoted as the optimal solution to (2). Under an additional assumption of geodesic convexity (Assumption 5.5), we can establish the following result that $\mathbb{E}\Delta'_t$ decreases strictly and can be bounded above.

**Theorem 5.15.** *Under assumptions 5.1, 5.2, 5.5, 5.6 and 5.7, suppose $\alpha \in (0, h_{max}^{-1}]$ and $\mu \in (0, L^{-1}]$. The sequence $\{J(\boldsymbol{w}'_t)\}_{t \geq s_o+1}$ satisfies the following relation:*

$$\mathbb{E}\Delta'_t \leq \frac{\zeta L D^2 + (t - s_o)\left(\frac{231\zeta\alpha\mu}{2\tau}G^2 + \frac{63\zeta\alpha\mu}{\tau}\sigma^2\right)}{2\zeta + t - s_o - 1}. \quad (31)$$

*Proof.* Appendix B.2. □

Theorem 5.15 establishes that non-asymptotic convergence of Algorithm 1 can be guaranteed after sufficient iterations

for sufficiently small step sizes $\mu$ and $\alpha$, if $J$ is geodesically convex and smooth.

Compared to the Euclidean counterpart (Chen & Sayed, 2012; Sayed et al., 2013; Vlaski & Sayed, 2021), key differences in our analysis include the impact of manifold curvature $\kappa$ (captured in the parameter $\zeta$) and the non-linear nature of the combination step (4). This makes traditional techniques like adjacency matrix decomposition unfeasible, since the network centroid cannot be computed using simple linear expressions. We address these challenges through a novel framework that studies network agreement via the evolution of the penalty term $P(\phi_t)$, and establish non-asymptotic convergence results using the carefully designed Lyapunov function in (30).

# 6. Examples and applications

In this section, we tailor Algorithm 1 to two common instances of Riemannian manifolds. The first one is the Grassmann manifold, a set of $k$-dimensional linear subspaces of $\mathbb{R}^p$, denoted by $\mathcal{G}_n^p$. The second is the manifold of $p \times p$ SPD matrices, denoted by $\mathcal{S}_n^{++}$. We consider applying our algorithms on $\mathcal{G}_n^p$ and a product manifold involving $\mathcal{S}_n^{++}$ to online distributed PCA and GMM inference, respectively. While the exponential map is convenient for theoretical analysis, the retractions often lead to more practical and efficient computations. Thus, for computational simplicity, we replace the exponential maps in the updates (3) and (4) with approximate retractions as in (Bonnabel, 2013). We provide definitions of the geodesic distance, Riemannian gradient, and retraction of $\mathcal{G}_n^p$ and $\mathcal{S}_n^{++}$ in Appendix C. The computational complexity of our algorithm is discussed in Appendix E.

## 6.1. Distributed PCA

We consider applying our algorithm on $\mathcal{G}_n^p$ to the online distributed PCA problem with $\boldsymbol{x}_k \in \mathbb{R}^n$ being data samples observed by each agent $k$. In the decentralized setting, we consider the following problem:

$$\min_{\pi(\boldsymbol{U}_k) \in \mathcal{G}_n^p} -\mathbb{E}_{\boldsymbol{x}_k}\left\{tr(\boldsymbol{U}_k^T \boldsymbol{x}_k \boldsymbol{x}_k^T \boldsymbol{U}_k)\right\}, \qquad (32)$$

where $\pi(\boldsymbol{U}_k)$ represents the local estimate at agent $k$. The expectation in the loss function (32) is approximated by realizations $\boldsymbol{x}_{k,t}$ at each time instant $t$. Note that although various works formulate PCA on the Stiefel manifold (Chen et al., 2021; Wang & Liu, 2022; Wang et al., 2023), the loss function in (32) is invariant to orthonormal transformations. Thus, we formulate the problem on the Grassmannian manifold since it makes the solution unique (Cunningham & Ghahramani, 2015). This formulation has also been found to have a similar mathematical structure of strong geodesic convexity, allowing arguments from convex optimization on

manifolds to be applied (Alimisis & Vandereycken, 2024). The Riemannian stochastic gradient of the loss function in (32) on $\mathcal{G}_n^p$ is computed using the Euclidean gradient of (32) at $\boldsymbol{U}_{k,t}$ and (90) given in Appendix C.1, leading to:

$$h(\boldsymbol{U}_{k,t}, \boldsymbol{x}_{k,t}) = 2(\boldsymbol{I} - \boldsymbol{U}_{k,t}\boldsymbol{U}_{k,t}^T)\boldsymbol{x}_{k,t}\boldsymbol{x}_{k,t}^T\boldsymbol{U}_{k,t}.$$

The retraction used is defined in (91). In order to evaluate the accuracy of the solutions, we consider the geodesic distance (89) between the estimates at each time instant $\pi(\boldsymbol{U}_{k,t})$ and the optimal solution $\pi(\boldsymbol{U}^*)$, and we define the mean square deviation (MSD) accordingly as

$$\text{MSD} = \frac{1}{K}\sum_{k=1}^K d_{\mathcal{G}_n^p}^2(\boldsymbol{U}_{k,t}, \boldsymbol{U}^*).$$

## 6.2. Distributed GMM inference

Another challenging application of our algorithm is distributed parameter estimation for GMMs with $\boldsymbol{x}_k \in \mathbb{R}^n$ being data samples observed by each agent $k$. The decentralized inference of mixtures of $M$ Gaussians with coefficients $\boldsymbol{\rho} \triangleq \{\rho_1, \cdots, \rho_M\}$, whose probability density is $p(\boldsymbol{x}) \triangleq \sum_{i=1}^M \rho_i \, p_{\mathcal{N}}(\boldsymbol{x}; \boldsymbol{m}_i, \boldsymbol{\Sigma}_i)$ with $p_{\mathcal{N}}$ a multivariate Gaussian with mean $\boldsymbol{m}_i$ and covariance $\boldsymbol{\Sigma}_i \succ 0$, can be reformulated as in (Hosseini & Sra, 2015):

$$\min_{\substack{\{\boldsymbol{S}_i\}_{i=1}^M \\ \{\eta_i\}_{i=1}^{M-1}}} -\mathbb{E}_{\boldsymbol{x}_k}\left\{\log\Big(\sum_{i=1}^M \frac{e^{\eta_i}}{\sum_{i=1}^M e^{\eta_i}} q_{\mathcal{N}}(\boldsymbol{y}_k; \boldsymbol{S}_i)\Big)\right\},$$
(33)

where $\boldsymbol{y}_k^T = [\boldsymbol{x}_k^T \; 1], \eta_i = \log\frac{\rho_i}{\rho_M}$ for $i = 1, \cdots, M-1$ and $\eta_M = 0$, which makes the problem unconstrained (Jordan & Jacobs, 1994), and $q_{\mathcal{N}}(\boldsymbol{y}_k; \boldsymbol{S}_i) = \sqrt{2\pi}e^{\frac{1}{2}}p_{\mathcal{N}}(\boldsymbol{y}_k; \boldsymbol{0}, \boldsymbol{S}_i)$. The problem (33) reformulated on the product manifold $\prod_{i=1}^M \mathcal{S}_n^{++} \times \mathbb{R}^{M-1}$ has the same optimum as that of the original log-likelihood of $p(\boldsymbol{x})$ (Hosseini & Sra, 2015), i.e.,

$$\boldsymbol{S}_i^* = \begin{pmatrix} \boldsymbol{\Sigma}_i^* + \boldsymbol{m}_i^*\boldsymbol{m}_i^{*T} & \boldsymbol{m}_i^* \\ \boldsymbol{m}_i^{*T} & 1 \end{pmatrix}.$$

The log-likelihood has been shown to be geodesically convex for the case of a single Gaussian (Hosseini & Sra, 2015), but not necessarily for multiple Gaussians. From this example, we can find the proposed algorithm itself can work even in some situations when not all these assumptions are satisfied. The Riemannian gradient of the loss function in (33) on the product manifold is composed of the (Riemannian) gradients w.r.t. $\{\boldsymbol{S}_i\}_{i=1}^M$ on $\prod_{i=1}^M \mathcal{S}_n^{++}$ and $\{\eta_i\}_{i=1}^{M-1}$ in $\mathbb{R}^{M-1}$. Specifically, the Riemannian gradient w.r.t. $\boldsymbol{S}_i$ was computed via the Euclidean gradient of (33) at $\boldsymbol{S}_{i,k,t}$ and (93) given in Appendix C.2. The retraction used is defined in (94). In this task, it is not very meaningful to compute the MSD values according to (92), because GMM

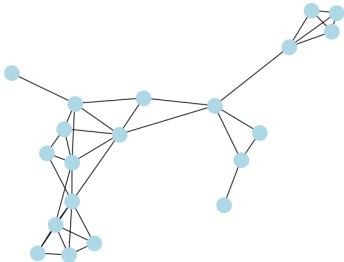

*Figure 1.* Graph topology.

is not inherently identifiable, which means that the parameters of the model may not be uniquely determined. Thus, to evaluate the performance of the solutions, we consider the average log-likelihood (ALL) as in (Hosseini & Sra, 2015; Collas et al., 2023).

# 7. Numerical experiments

In this section, we present numerical experiments on distributed PCA and parameter estimation of GMMs, which are formulated on manifolds as explained in Section 6. Our method is implemented in Python with the Pymanopt toolbox (Townsend et al., 2016). Open-source code to reproduce the results is publicly available on `https://github.com/xiuheng-wang/diffusion_manifold_release`. The graph topology of the multi-agent system used for the experiments is illustrated in Figure 1. The weights in matrix $C$ were randomly generated by the Metropolis rule (Xiao et al., 2006) with $K = 20$ agents[3]. For simulation on synthetic data, the MSD results are averaged over 100 times independent Monte Carlo experiments. Hereafter, we briefly describe the baselines.

**Baselines:** We compare our algorithm against the Riemannian non-cooperative and centralized strategies for both PCA and GMM inference. The non-cooperative algorithm independently applies R-SGD on each agent using its local data $x_{k,t}$, while the centralized works on data $X_t = \{x_{k,t}\}_{k=1}^K$ collected from all agents. We also provide comparisons with an extrinsic consensus algorithm on the Stiefel manifold: Decentralized Riemannian Stochastic Gradient Descent (DRSGD) (Chen et al., 2021) for PCA. For GMM inference, to the best of our knowledge there are no approaches that are both online and decentralized. Thus, for comparison, we extend the decentralized consensus SGD (Nedic et al., 2010; Lian et al., 2017) to the product manifold presented in Section 6.2 using a projection operator to ensure the constraints are satisfied. This Extrinsic Consensus strategy for GMM inference is named ECGMM.

---

[3]To illustrate the applicability to more networks, we include additional experimental results in Appendix D.2.

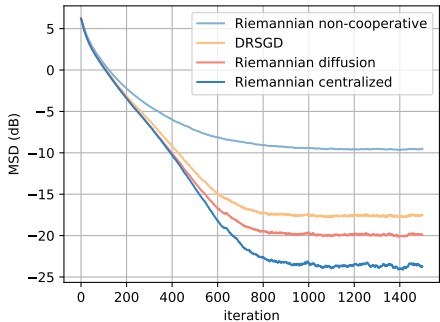

*Figure 2.* Illustration of MSD performance of the algorithms for distributed PCA on synthetic data.

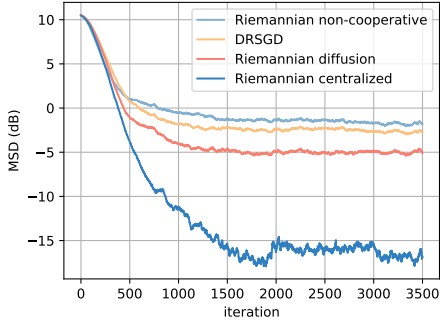

*Figure 3.* Illustration of MSD performance of the algorithms for distributed PCA on real data.

## 7.1. Experiments on PCA

We first present results on PCA formulated on $\mathcal{G}_n^p$ with both synthetic and real data.

**Synthetic data:** We generate synthetic data as in (Chen et al., 2021). First, we set $n = 10$, $p = 5$, and independently sample $1500K$ data points according to a multivariate Gaussian model to obtain a matrix $S \in \mathbb{R}^{n \times 1500K}$. Let $S = U\Lambda V^T$ be its truncated SVD. We modify the distribution of $\Lambda$ as $\Lambda' = \text{diag}(\lambda^i)$ with $\lambda = 0.8$ and $i = 0, \cdots, n-1$ to reset $S$ as $S' = U\Lambda'V^T$. We randomly shuffle and split the columns of $S' \in \mathbb{R}^{n \times 1500K}$ into 1500 subsets to obtain $X_t$ for all time instants $t = 1, \ldots, 1500$. The simulations used fixed step sizes $\mu = 0.05$ and $\alpha = 0.8$. For our algorithm, the step sizes control the tradeoff between convergence speed and steady-state performance; this is illustrated with experimental results in Appendix D.3.

**Real data:** We also obtain numerical results on the MNIST dataset (LeCun, 1998). The dataset contains 70000 hand-written images with $n = 784$ pixels. The data matrix is normalized such that the elements are in the range $[0, 1]$ and then centered. To compute MSD, we perform PCA on the full data matrix and regard its result as the optimum. We randomly shuffle the images, partition them into $K = 20$ subsets, and then run the algorithms to compute the first $p = 5$ principal components with the fixed step sizes $\mu = 0.002$ and $\alpha = 0.005$.

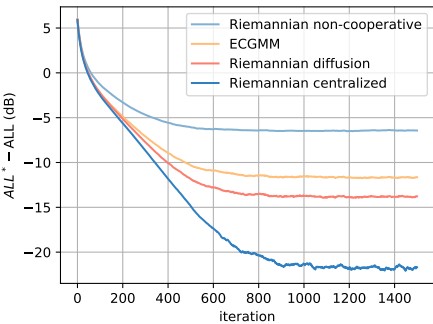

*Figure 4.* Illustration of ALL differences of the algorithms for distributed GMM inference on synthetic data.

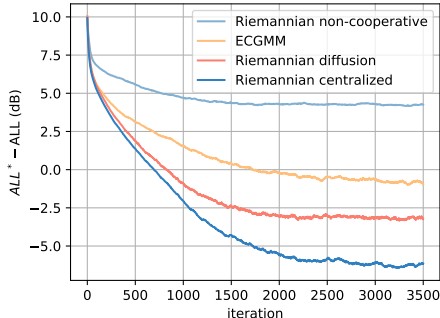

*Figure 5.* Illustration of ALL differences of the algorithms for distributed GMM inference on real data.

**Discussion:** Figure 2 shows the MSD learning curves for the compared algorithms on synthetic data. It can be seen that the Riemannian diffusion adaptation strategy achieves a significant improvement in MSD performance compared to the non-cooperative case, which indicates the benefit of information exchange. Moreover, our method also outperforms DRSGD. The centralized solution achieves the lowest MSD, as it can access information over the whole graph. The proposed algorithm is fully decentralized, where each agent uses only locally observed data to update its local estimate and exchange information only among neighboring agents. Although the proposed algorithm has lower performance compared to the centralized method, it can be computed in parallel on multiple agents. The MSD of the different methods on real data, shown in Figure 3, behaves similarly to that in the experiment with synthetic data, showing the same comparative performances between the different approaches.

### 7.2. Experiments on GMM inference

Now we show results on a more challenging task: GMM inference formulated on $\prod_{i=1}^{M} \mathcal{S}_n^{++} \times \mathbb{R}^{M-1}$ with synthetic and real data. As in (Hosseini & Sra, 2015), we initialize the mixture parameters for all the methods using k-means++.

**Synthetic data:** To generate synthetic data, we choose the parameters $\boldsymbol{m}, \boldsymbol{\Sigma}$ and $\boldsymbol{\rho}$ of the Gaussian mixture similarly to (Collas et al., 2023). First, $\boldsymbol{\Sigma}$ is generated using its eigen-decomposition $\boldsymbol{\Sigma} = \boldsymbol{U}\boldsymbol{\Lambda}\boldsymbol{U}^T$. $\boldsymbol{U}$ is drawn from the uniform distribution on the orthogonal group, and all the elements of $\boldsymbol{\Lambda}$ are drawn from a chi-squared distribution with a degree of freedom 3. Second, the elements of $\boldsymbol{\rho}$ are drawn from a Gamma distribution with a shape parameter 10, and then normalized. Third, $\boldsymbol{m}$ is sampled from a multivariate Gaussian distribution $\mathcal{N}(\boldsymbol{0}, 3\boldsymbol{I})$, where sampling is repeated until the following inequality is satisfied (Hosseini & Sra, 2015):

$$\forall i, j, \ \|\boldsymbol{m}_i - \boldsymbol{m}_j\| \geq \max_{i,j}\{tr(\boldsymbol{\Sigma}_i), tr(\boldsymbol{\Sigma}_j)\}.$$

We set $n = 8$ and $M = 3$ and independently sample $1500K$ data points according to the Gaussian mixture above. These

data points are randomly shuffled and split into $1500$ subsets to obtain $\boldsymbol{X}_t$ for all time instants $t = 1, \ldots, 1500$. The simulations used fixed step sizes $\mu = 0.04$ and $\alpha = 0.05$.

**Real data:** Again, we perform the same data processing for the MNIST dataset (LeCun, 1998) as in Section 7.1. Then, we apply PCA to reduce the dimensionality $n = 20$. We compute the Expectation Maximization (EM) solution on the full dataset and regard its result as an optimum. To evaluate the performance, we compare the difference between the ALL values of the optimum and estimated solutions, denoted as $\text{ALL}^* - \text{ALL}$. We implement the compared algorithms to infer mixtures of Gaussian models with $M = 7$ and fixed step sizes $\mu = 0.08$ and $\alpha = 0.08$.

**Discussion:** Figure 4 and Figure 5 illustrate the ALL difference values for the compared methods on synthetic and real data, respectively. The results demonstrate that our method outperforms both the non-cooperative algorithm and ECGMM, further highlighting its effectiveness. As expected, the centralized case achieves the lowest ALL difference.

## 8. Conclusions

In this paper, the Riemannian diffusion adaptation algorithm is proposed. The strategy consists of two efficient steps: an adaptation step, where R-SGD is used at each agent to update the estimate of the local solution on the manifold, and a combination step, where the estimates of neighboring agents are combined on the tangent space. A theoretical analysis is provided under constant step size, showing that network agreement is achieved with high probability and the algorithm converges non-asymptotically to a neighborhood of the optimal solution. The proposed method is applied to online decentralized PCA and GMM inference. Experimental results on both synthetic and real-world data illustrate the efficacy of the proposed strategy. One main limitation of this work is that the theoretical results rely on the use of the exponential map, which can be computationally heavy. This is discussed in more detail in Appendix F.

## Impact Statement

This paper presents work that aims to advance the field of Machine Learning. There are many potential societal consequences of our work, none of which we feel must be specifically highlighted here.

## Acknowledgements

The authors would like to thank the reviewers for their constructive feedback. The work of Cédric Richard was supported in part by the French Government through the 3IA Côte d'Azur Investments in the Future Project under grant ANR-19-P3IA-0002, and in part by grant ANR-19-CE48-0002. The work of Ricardo Borsoi was supported in part by the French National Research Agency, under grants ANR-23-CE23-0024, ANR-23-CE94-0001, and by the National Science Foundation, under grant NSF 2316420. Xiuheng Wang would like to thank Dr. Mengfei Zhang for the beneficial discussion in the early exploratory stage of this work.

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

# A. Proofs of the lemmas

### A.1. Lemma 5.8

From the definition of $P(\boldsymbol{\phi}_t)$, we have

$$\|\nabla P(\boldsymbol{\phi}_t)\|^2 = \sum_{k=1}^{K} \left\| -\sum_{\ell=1}^{K} c_{\ell k} \exp_{\boldsymbol{\phi}_{k,t}}^{-1}(\boldsymbol{\phi}_{\ell,t}) \right\|^2 \leq \sum_{k=1}^{K} \sum_{\ell=1}^{K} c_{\ell k} \left\| \exp_{\boldsymbol{\phi}_{k,t}}^{-1}(\boldsymbol{\phi}_{\ell,t}) \right\|^2 = 2P(\boldsymbol{\phi}_t), \tag{34}$$

where the inequality follows from Jensen's inequality and the fact that $C$ is left-stochastic, as stated in Assumption 5.2.

### A.2. Lemma 5.9

Due to the g-convexity of $J_k$ under Assumption 5.5, we have

$$J_k(\boldsymbol{w}_m) - J_k(\boldsymbol{w}_k) \leq \langle -\nabla J_k(\boldsymbol{w}_m), \exp_{\boldsymbol{w}_m}^{-1}(\boldsymbol{w}_k) \rangle \tag{35}$$

Multiply the above by $\frac{1}{K}$ and sum over $k$, we get

$$\frac{1}{K} \sum_k J_k(\boldsymbol{w}_m) - \frac{1}{K} \sum_k J_k(\boldsymbol{w}_k) \leq \frac{1}{K} \sum_k \langle -\nabla J_k(\boldsymbol{w}_m), \exp_{\boldsymbol{w}_m}^{-1}(\boldsymbol{w}_k) \rangle \tag{36}$$

Using $\frac{1}{K} \sum_k J_k(\boldsymbol{w}_m) := J(\bar{\boldsymbol{w}})$ and rearranging the terms, this can be rewritten as

$$J(\bar{\boldsymbol{w}}) - \frac{1}{K} \sum_k J_k(\boldsymbol{w}_k) \leq \langle -\nabla J_k(\boldsymbol{w}_m), \frac{1}{K} \sum_k \exp_{\boldsymbol{w}_m}^{-1}(\boldsymbol{w}_k) \rangle \tag{37}$$

Note that $\frac{1}{K} \sum_k \exp_{\boldsymbol{w}_m}^{-1}(\boldsymbol{w}_k) = \frac{1}{K} \nabla_{\boldsymbol{x}} \sum_k d^2(\boldsymbol{w}_k, \boldsymbol{x})\big|_{\boldsymbol{x}:=\boldsymbol{w}_m}$ is the gradient of the cost function $\frac{1}{K} \sum_k d^2(\boldsymbol{w}_k, \boldsymbol{x})$ evaluated at the Fréchet mean $\boldsymbol{w}_m$, which is its minimizer, therefore, the first order optimality condition implies $\frac{1}{K} \sum_k \exp_{\boldsymbol{w}_m}^{-1}(\boldsymbol{w}_k) = 0$. Combining this with the previous results leads to $J(\bar{\boldsymbol{w}}) \leq \frac{1}{K} \sum_k J_k(\boldsymbol{w}_k) = J(\boldsymbol{w})$.

### A.3. Lemma 5.10

Since $\mathcal{B}$ is compact (Assumption 5.1), from the geodesic smoothness of $J_k$ (Assumption 5.6), we have:

$$\|\nabla J_k(\boldsymbol{w}_t)\| \leq G, \tag{38}$$

for a non-negative constant $G < \infty$. Observe that (38) implies a similar condition on the deviation from the centralized gradient via Jensen's inequality:

$$\|\nabla J(\boldsymbol{w}_t)\| = \left\| \frac{1}{K} \sum_k \nabla J_k(\boldsymbol{w}_t) \right\| \leq \frac{1}{K} \sum_k \|\nabla J_k(\boldsymbol{w}_t)\| \leq G. \tag{39}$$

### A.4. Lemma 5.11

Let us start with the update $\boldsymbol{w}_t = \exp_{\boldsymbol{\phi}_t}\big( -\alpha \nabla P(\boldsymbol{\phi}_t) \big)$ from (7), define $\gamma_1(\alpha) \triangleq \exp_{\boldsymbol{\phi}_t}\big( -\alpha \nabla P(\boldsymbol{\phi}_t) \big)$ as the minimal geodesic from $\boldsymbol{\phi}_t$ to $\boldsymbol{w}_t$, and use the second-order Taylor expansion of $\alpha \mapsto P(\gamma_1(\alpha))$ around $\alpha = 0$, under assumptions 5.1 and 5.2, then we have (Tron et al., 2012)

$$P(\boldsymbol{w}_t) \leq P(\boldsymbol{\phi}_t) + \langle \nabla P(\boldsymbol{\phi}_t), -\alpha \nabla P(\boldsymbol{\phi}_t) \rangle + \frac{h_{max}\| -\alpha \nabla P(\boldsymbol{\phi}_t)\|^2}{2}$$
$$= P(\boldsymbol{\phi}_t) - \epsilon \|\nabla P(\boldsymbol{\phi}_t)\|^2, \tag{40}$$

where $\epsilon \triangleq \alpha\big(1 - \frac{\alpha h_{max}}{2}\big) > 0$ since $\alpha \in (0, h_{max}^{-1}]$. Also, we use the first-order Taylor expansion of $\alpha \mapsto \nabla P(\gamma_1(\alpha))$ around $\alpha = 0$ to obtain the following bound:

$$\|\nabla P(\boldsymbol{w}_t) - \Gamma_{\boldsymbol{\phi}_t}^{\boldsymbol{w}_t} \nabla P(\boldsymbol{\phi}_t)\| \leq h_{max}\alpha \|\nabla P(\boldsymbol{\phi}_t)\|. \tag{41}$$

Similarly, for the update $\phi_{t+1} = \exp_{\boldsymbol{w}_t}\left(-\mu\widehat{\nabla J}(\boldsymbol{w}_t)\right)$ from (6), define $\gamma_2(\mu) \triangleq \exp_{\boldsymbol{w}_t}\left(-\mu\widehat{\nabla J}(\boldsymbol{w}_t)\right)$ as the minimal geodesic from $\boldsymbol{w}_t$ to $\phi_{t+1}$, use the second-order Taylor expansion of $\mu \mapsto P(\gamma_2(\mu))$ around $\mu = 0$, under assumptions 5.1 and 5.2, then we have

$$P(\phi_{t+1}) \leq P(\boldsymbol{w}_t) + \langle \nabla P(\boldsymbol{w}_t), -\mu\widehat{\nabla J}(\boldsymbol{w}_t)\rangle + \frac{h_{max}\mathbb{E}\| -\mu\widehat{\nabla J}(\boldsymbol{w}_t)\|^2}{2} . \tag{42}$$

Take the expectation on (42) w.r.t. $\{\boldsymbol{x}_s\}_{s=0}^t$ and consider (17) and (18) in Assumption 5.7, we have

$$\begin{aligned}
\mathbb{E}P(\phi_{t+1}) &\leq \mathbb{E}P(\boldsymbol{w}_t) + \mathbb{E}\{\langle \nabla P(\boldsymbol{w}_t), -\mu\widehat{\nabla J}(\boldsymbol{w}_t)\rangle\} + \frac{h_{max}\mathbb{E}\| -\mu\widehat{\nabla J}(\boldsymbol{w}_t)\|^2}{2} \\
&= \mathbb{E}P(\boldsymbol{w}_t) + \mathbb{E}\{\langle \nabla P(\boldsymbol{w}_t), -\mu\mathbb{E}\{\widehat{\nabla J}(\boldsymbol{w}_t)|\mathcal{F}_t\}\rangle\} + \frac{h_{max}\mu^2}{2}\mathbb{E}\|\widehat{\nabla J}(\boldsymbol{w}_t)\|^2 \\
&= \mathbb{E}P(\boldsymbol{w}_t) + \mathbb{E}\{\langle \nabla P(\boldsymbol{w}_t), -\mu\nabla J(\boldsymbol{w}_t)\rangle\} + \frac{h_{max}\mu^2}{2}\mathbb{E}\|\widehat{\nabla J}(\boldsymbol{w}_t)\|^2 \\
&\leq \mathbb{E}P(\boldsymbol{w}_t) + \frac{\xi}{2}\mathbb{E}\|\nabla P(\boldsymbol{w}_t)\|^2 + \frac{1}{2\xi}\mu^2\mathbb{E}\|\nabla J(\boldsymbol{w}_t)\|^2 + \frac{h_{max}\mu^2}{2}\mathbb{E}\|\widehat{\nabla J}(\boldsymbol{w}_t) - \nabla J(\boldsymbol{w}_t) + \nabla J(\boldsymbol{w}_t)\|^2 \\
&\leq \mathbb{E}P(\boldsymbol{w}_t) + \frac{\xi}{2}\mathbb{E}\|\nabla P(\boldsymbol{w}_t)\|^2 + \left(\frac{1}{2\xi} + h_{max}\right)\mu^2\mathbb{E}\|\nabla J(\boldsymbol{w}_t)\|^2 + h_{max}\mu^2\mathbb{E}\{\mathbb{E}\{\|\widehat{\nabla J}(\boldsymbol{w}_t) - \nabla J(\boldsymbol{w}_t)\|^2|\mathcal{F}_t\}\} \\
&= \mathbb{E}P(\boldsymbol{w}_t) + \frac{\xi}{2}\mathbb{E}\|\nabla P(\boldsymbol{w}_t)\|^2 + \left(\frac{1}{2\xi} + h_{max}\right)\mu^2\mathbb{E}\|\nabla J(\boldsymbol{w}_t)\|^2 + h_{max}\mu^2\sigma^2 ,
\end{aligned} \tag{43}$$

where we use the facts $\langle a, b\rangle \leq \frac{\xi}{2}a^2 + \frac{1}{2\xi}b^2$ for $\xi > 0$ and $\frac{1}{2}(a+b)^2 \leq a^2 + b^2$ in the second and third inequalities, respectively. Next, we take the expectation on (40) w.r.t. $\{\boldsymbol{x}_s\}_{s=0}^t$, and combine the result with (43) to obtain

$$\mathbb{E}P(\phi_{t+1}) \leq \mathbb{E}P(\phi_t) - \epsilon\mathbb{E}\|\nabla P(\phi_t)\|^2 + \frac{\xi}{2}\mathbb{E}\|\nabla P(\boldsymbol{w}_t)\|^2 + \left(\frac{1}{2\xi} + h_{max}\right)\mu^2\mathbb{E}\|\nabla J(\boldsymbol{w}_t)\|^2 + h_{max}\mu^2\sigma^2. \tag{44}$$

Now we need to upper bound $\mathbb{E}\|\nabla P(\boldsymbol{w}_t)\|^2$. Consider

$$\begin{aligned}
\frac{1}{2}\mathbb{E}\|\nabla P(\boldsymbol{w}_t)\|^2 &= \frac{1}{2}\mathbb{E}\|\nabla P(\boldsymbol{w}_t) - \Gamma_{\phi_t}^{\boldsymbol{w}_t}\nabla P(\phi_t) + \Gamma_{\phi_t}^{\boldsymbol{w}_t}\nabla P(\phi_t)\|^2 \\
&\leq \mathbb{E}\|\nabla P(\boldsymbol{w}_t) - \Gamma_{\phi_t}^{\boldsymbol{w}_t}\nabla P(\phi_t)\|^2 + \mathbb{E}\|\nabla P(\phi_t)\|^2 \\
&\leq (\alpha^2 h_{max}^2 + 1)\mathbb{E}\|\nabla P(\phi_t)\|^2 ,
\end{aligned} \tag{45}$$

where the first inequality uses the fact $\frac{1}{2}(a+b)^2 \leq a^2 + b^2$, the second inequality uses (41). Plugging the upper bound of $\frac{1}{2}\mathbb{E}\|\nabla P(\boldsymbol{w}_t)\|^2$, as provided in (45), into (44) and reordering, we have

$$\begin{aligned}
\mathbb{E}P(\phi_{t+1}) &\leq \mathbb{E}P(\phi_t) - \epsilon\mathbb{E}\|\nabla P(\phi_t)\|^2 + \xi(\alpha^2 h_{max}^2 + 1)\mathbb{E}\|\nabla P(\phi_t)\|^2 + \left(\frac{1}{2\xi} + h_{max}\right)\mu^2\mathbb{E}\|\nabla J(\boldsymbol{w}_t)\|^2 + h_{max}\mu^2\sigma^2 \\
&= \mathbb{E}P(\phi_t) - \frac{\epsilon}{2}\mathbb{E}\|\nabla P(\phi_t)\|^2 + \left(\frac{\alpha^2 h_{max}^2 + 1}{\epsilon} + h_{max}\right)\mu^2\mathbb{E}\|\nabla J(\boldsymbol{w}_t)\|^2 + h_{max}\mu^2\sigma^2 \\
&\leq \mathbb{E}P(\phi_t) - \frac{\epsilon}{2}\mathbb{E}\|\nabla P(\phi_t)\|^2 + \left(\frac{\alpha^2 h_{max}^2 + 1}{\epsilon} + h_{max}\right)\mu^2 G^2 + h_{max}\mu^2\sigma^2 ,
\end{aligned} \tag{46}$$

where in the equality we select $\xi = \frac{\epsilon}{2(\alpha^2 h_{max}^2 + 1)}$ for simplicity, and in the second inequality we use (21) from Lemma 5.10. Since $\alpha \in (0, h_{max}^{-1}]$, we have $h_{max} \leq \alpha^{-1}$ and $\epsilon \geq \frac{\alpha}{2}$, and thus we can further simplify (46) as

$$\mathbb{E}P(\phi_{t+1}) \leq \mathbb{E}P(\phi_t) - \frac{\alpha}{4}\mathbb{E}\|\nabla P(\phi_t)\|^2 + \frac{5\mu^2}{\alpha}G^2 + \frac{\mu^2}{\alpha}\sigma^2 . \tag{47}$$

Re-arranging the terms in (47) gives the desired result.

## A.5. Lemma 5.14

Consider the smoothness property of $J$ in Assumption 5.6 with $\exp_{\boldsymbol{w}_t}^{-1}(\boldsymbol{\phi}_{t+1}) = -\mu\widehat{\nabla J}(\boldsymbol{w}_t)$ from (6), we can write:

$$J(\boldsymbol{\phi}_{t+1}) \leq J(\boldsymbol{w}_t) + \langle \nabla J(\boldsymbol{w}_t), \exp_{\boldsymbol{w}_t}^{-1}(\boldsymbol{\phi}_{t+1})\rangle + \frac{L\|\exp_{\boldsymbol{w}_t}^{-1}(\boldsymbol{\phi}_{t+1})\|^2}{2}$$

$$= J(\boldsymbol{w}_t) + \langle \nabla J(\boldsymbol{w}_t), -\mu\widehat{\nabla J}(\boldsymbol{w}_t)\rangle + \frac{L\|-\mu\widehat{\nabla J}(\boldsymbol{w}_t)\|^2}{2}. \tag{48}$$

Also, we can obtain the following bound:

$$\|\nabla J(\boldsymbol{\phi}_{t+1}) - \Gamma_{\boldsymbol{w}_t}^{\boldsymbol{\phi}_{t+1}}\nabla J(\boldsymbol{w}_t)\| \leq L\mu\|\widehat{\nabla J}(\boldsymbol{w}_t)\|. \tag{49}$$

Take expectation on (48) w.r.t. $\{\boldsymbol{x}_s\}_{s=0}^t$ and consider (17) in Assumption 5.7, we have:

$$\mathbb{E}J(\boldsymbol{\phi}_{t+1}) \leq \mathbb{E}J(\boldsymbol{w}_t) + \mathbb{E}\{\langle \nabla J(\boldsymbol{w}_t), -\mu\widehat{\nabla J}(\boldsymbol{w}_t)\rangle\} + \frac{L\mathbb{E}\|-\mu\widehat{\nabla J}(\boldsymbol{w}_t)\|^2}{2}$$

$$= \mathbb{E}J(\boldsymbol{w}_t) + \mathbb{E}\{\langle \mathbb{E}\{\widehat{\nabla J}(\boldsymbol{w}_t)|\mathcal{F}_t\}, -\mu\widehat{\nabla J}(\boldsymbol{w}_t)\rangle\} + \frac{L\mu^2}{2}\mathbb{E}\|\widehat{\nabla J}(\boldsymbol{w}_t)\|^2$$

$$= \mathbb{E}J(\boldsymbol{w}_t) - \epsilon\mathbb{E}\|\widehat{\nabla J}(\boldsymbol{w}_t)\|^2. \tag{50}$$

where $\epsilon \triangleq \mu\left(1 - \frac{\mu L}{2}\right) > 0$ since $\mu \in (0, L^{-1}]$. Again, consider the smoothness property of $J$ in Assumption 5.6 with $\exp_{\boldsymbol{\phi}_{t+1}}^{-1}(\boldsymbol{w}_{t+1}) = -\alpha\nabla P(\boldsymbol{\phi}_{t+1})$ from (7), we obtain:

$$J(\boldsymbol{w}_{t+1}) \leq J(\boldsymbol{\phi}_{t+1}) + \langle \nabla J(\boldsymbol{\phi}_{t+1}), \exp_{\boldsymbol{\phi}_{t+1}}^{-1}(\boldsymbol{w}_{t+1})\rangle + \frac{L\|\exp_{\boldsymbol{\phi}_{t+1}}^{-1}(\boldsymbol{w}_{t+1})\|^2}{2}$$

$$= J(\boldsymbol{\phi}_{t+1}) + \langle \nabla J(\boldsymbol{\phi}_{t+1}), -\alpha\nabla P(\boldsymbol{\phi}_{t+1})\rangle + \frac{L\|-\alpha\nabla P(\boldsymbol{\phi}_{t+1})\|^2}{2}$$

$$\leq J(\boldsymbol{\phi}_{t+1}) + \frac{\xi}{2}\|\nabla J(\boldsymbol{\phi}_{t+1})\|^2 + \left(\frac{1}{2\xi} + L\right)\alpha^2\|\nabla P(\boldsymbol{\phi}_{t+1})\|^2, \tag{51}$$

where the second inequality uses the fact $\langle a, b\rangle \leq \frac{\xi}{2}a^2 + \frac{1}{2\xi}b^2$. Next, we take the expectation on (51) w.r.t. $\{\boldsymbol{x}_s\}_{s=0}^t$, and combine the result with (50) to obtain

$$\mathbb{E}J(\boldsymbol{w}_{t+1}) \leq \mathbb{E}J(\boldsymbol{w}_t) - \epsilon\mathbb{E}\|\widehat{\nabla J}(\boldsymbol{w}_t)\|^2 + \frac{\xi}{2}\mathbb{E}\|\nabla J(\boldsymbol{\phi}_{t+1})\|^2 + \left(\frac{1}{2\xi} + L\right)\alpha^2\mathbb{E}\|\nabla P(\boldsymbol{\phi}_{t+1})\|^2, \tag{52}$$

Now we need to upper bound $\mathbb{E}\|\nabla J(\boldsymbol{\phi}_{t+1})\|^2$. Consider

$$\frac{1}{2}\mathbb{E}\|\nabla J(\boldsymbol{\phi}_{t+1})\|^2 = \frac{1}{2}\mathbb{E}\|\nabla J(\boldsymbol{\phi}_{t+1}) - \Gamma_{\boldsymbol{w}_t}^{\boldsymbol{\phi}_{t+1}}\nabla J(\boldsymbol{w}_t) + \Gamma_{\boldsymbol{w}_t}^{\boldsymbol{\phi}_{t+1}}\nabla J(\boldsymbol{w}_t)\|^2$$

$$\leq \mathbb{E}\|\nabla J(\boldsymbol{\phi}_{t+1}) - \Gamma_{\boldsymbol{w}_t}^{\boldsymbol{\phi}_{t+1}}\nabla J(\boldsymbol{w}_t)\|^2 + \mathbb{E}\|\nabla J(\boldsymbol{w}_t)\|^2$$

$$\leq (\mu^2 L^2 + 1)\mathbb{E}\|\widehat{\nabla J}(\boldsymbol{w}_t)\|^2, \tag{53}$$

where the first inequality uses the fact $\frac{1}{2}(a + b)^2 \leq a^2 + b^2$, the second inequality uses (49) and the fact $\mathbb{E}\|\nabla J(\boldsymbol{w}_t)\|^2 \leq \mathbb{E}\|\widehat{\nabla J}(\boldsymbol{w}_t)\|^2$. Plugging the upper bound of $\frac{1}{2}\mathbb{E}\|\nabla J(\boldsymbol{\phi}_{t+1})\|^2$, as provided in (53), into (52) and reordering, we have

$$\mathbb{E}J(\boldsymbol{w}_{t+1}) \leq \mathbb{E}J(\boldsymbol{w}_t) - \left(\epsilon - \xi(\mu^2 L^2 + 1)\right)\mathbb{E}\|\widehat{\nabla J}(\boldsymbol{w}_t)\|^2 + \left(\frac{1}{2\xi} + L\right)\alpha^2\mathbb{E}\|\nabla P(\boldsymbol{\phi}_{t+1})\|^2$$

$$= \mathbb{E}J(\boldsymbol{w}_t) - \frac{\epsilon}{2}\mathbb{E}\|\widehat{\nabla J}(\boldsymbol{w}_t)\|^2 + \left(\frac{\mu^2 L^2 + 1}{\epsilon} + L\right)\alpha^2\mathbb{E}\|\nabla P(\boldsymbol{\phi}_{t+1})\|^2, \tag{54}$$

where in the equality we select $\xi = \frac{\epsilon}{2(\mu^2 L^2 + 1)}$ for simplicity. Since $\mu \in (0, L^{-1}]$, we have $L \leq \mu^{-1}$ and $\epsilon \geq \frac{\mu}{2}$, and thus we can further simplify (54) as

$$\mathbb{E}J(\boldsymbol{w}_{t+1}) \leq \mathbb{E}J(\boldsymbol{w}_t) - \frac{\mu}{4}\mathbb{E}\|\widehat{\nabla J}(\boldsymbol{w}_t)\|^2 + \frac{5\alpha^2}{\mu}\mathbb{E}\|\nabla P(\boldsymbol{\phi}_{t+1})\|^2. \tag{55}$$

Re-arranging the terms in (55) gives the desired result.

## B. Proofs of the theorems

### B.1. Theorem 5.12

Define $\bar{\phi}_t = \text{col}\{\phi_{m,t}, \cdots, \phi_{m,t}\}$ with $\phi_{m,t}$ being the Fréchet mean of $\phi_t$, i.e., $\phi_{m,t} \triangleq \arg\min_\phi \sum_{k=1}^K d^2(\phi_{k,t}, \phi)$. Further, define $\gamma_3(\beta) \triangleq \exp_{\phi_t}\left(\beta \exp_{\phi_t}^{-1} \bar{\phi}_t\right)$ as the minimal geodesic from $\phi_t$ to $\bar{\phi}_t$, use the second-order Taylor expansion of $\beta \mapsto P(\gamma_3(\beta))$ around $\beta = 0$, then we have (Afsari et al., 2013):

$$\langle \nabla P(\phi_t), \exp_{\phi_t}^{-1}(\bar{\phi}_t)\rangle + \frac{h_{min}\|\exp_{\phi_t}^{-1}(\bar{\phi}_t)\|^2}{2} \le P(\bar{\phi}_t) - P(\phi_t), \tag{56}$$

Considering $P(\bar{\phi}_t) = 0$, we can further write

$$\begin{aligned}
P(\phi_t) = P(\phi_t) - P(\bar{\phi}_t) &\le \langle -\nabla P(\phi_t), \exp_{\phi_t}^{-1}(\bar{\phi}_t)\rangle - \frac{h_{min}\|\exp_{\phi_t}^{-1}(\bar{\phi}_t)\|^2}{2} \\
&\le \frac{1 - \alpha h_{min}}{2\alpha} d^2(\phi_t, \bar{\phi}_t) - \frac{1}{2\alpha} d^2(w_t, \bar{\phi}_t) + \frac{\zeta\alpha}{2}\|\nabla P(\phi_t)\|^2 \\
&\le \frac{1 - \alpha h_{min}}{2\alpha} d^2(\phi_t, \bar{\phi}_t) - \frac{1}{2\alpha} d^2(w_t, \bar{w}_t) + \frac{\zeta\alpha}{2}\|\nabla P(\phi_t)\|^2,
\end{aligned} \tag{57}$$

where the second inequality is from Corollary 5.4, for the update $w_t = \exp_{\phi_t}\left(-\alpha\nabla P(\phi_t)\right)$ from (7) and the third inequality uses the fact $d^2(w_t, \bar{w}_t) = \sum_{k=1}^K d^2(w_{k,t}, w_{c,t}) \le \sum_{k=1}^K d^2(w_{k,t}, \phi_{m,t}) = d^2(w_t, \bar{\phi}_t)$ where $\bar{w}_t = \text{col}\{w_{m,t}, \cdots, w_{m,t}\}$ with $w_{m,t}$ being the Fréchet mean of $w_t$.

From Corollary 5.4, for the update $\phi_{t+1} = \exp_{w_t}\left(-\mu\widehat{\nabla J}(w_t)\right)$ in (6), we have

$$\langle -\widehat{\nabla J}(w_t), \exp_{w_t}^{-1}(\bar{w}_t)\rangle \le \frac{1}{2\mu} d^2(w_t, \bar{w}_t) - \frac{1}{2\mu} d^2(\phi_{t+1}, \bar{w}_t) + \frac{\zeta\mu}{2}\|\widehat{\nabla J}(w_t)\|^2. \tag{58}$$

Take the expectation on the previous result w.r.t. $\{x_s\}_{s=0}^t$ and consider (17) in Assumption 5.7, we obtain

$$\begin{aligned}
\mathbb{E}\{\langle -\widehat{\nabla J}(w_t), \exp_{w_t}^{-1}(\bar{w}_t)\rangle\} &= \mathbb{E}\{\langle -\mathbb{E}\{\widehat{\nabla J}(w_t)|\mathcal{F}_t\}, \exp_{w_t}^{-1}(\bar{w}_t)\rangle\} \\
&= \mathbb{E}\{\langle -\nabla J(w_t), \exp_{w_t}^{-1}(\bar{w}_t)\rangle\},
\end{aligned} \tag{59}$$

Combining (58) and (59), we can write

$$\mathbb{E}\{\langle -\nabla J(w_t), \exp_{w_t}^{-1}(\bar{w}_t)\rangle\} \le \frac{1}{2\mu}\mathbb{E}d^2(w_t, \bar{w}_t) - \frac{1}{2\mu}\mathbb{E}d^2(\phi_{t+1}, \bar{w}_t) + \frac{\zeta\mu}{2}\mathbb{E}\|\widehat{\nabla J}(w_t)\|^2, \tag{60}$$

Consider $J$ to be a geodesically convex function under Assumption 5.5. Using (13), taking its expectation w.r.t. $\{x_s\}_{s=0}^t$ and combining the result with (60), we further write

$$\begin{aligned}
\mathbb{E}\{J(w_t) - J(\bar{w}_t)\} &\le \mathbb{E}\{\langle -\nabla J(w_t), \exp_{w_t}^{-1}(\bar{w}_t)\rangle\} \\
&\le \frac{1}{2\mu}\mathbb{E}d^2(w_t, \bar{w}_t) - \frac{1}{2\mu}\mathbb{E}d^2(\phi_{t+1}, \bar{w}_t) + \frac{\zeta\mu}{2}\mathbb{E}\|\widehat{\nabla J}(w_t)\|^2.
\end{aligned} \tag{61}$$

Using $J(\bar{w}_t) \le J(w_t)$ in Lemma 5.9, from (61), we have

$$\begin{aligned}
-\mathbb{E}d^2(w_t, \bar{w}_t) &\le -\mathbb{E}d^2(\phi_{t+1}, \bar{\phi}_{t+1}) + \zeta\mu^2\mathbb{E}\|\widehat{\nabla J}(w_t)\|^2 \\
&\le -\mathbb{E}d^2(\phi_{t+1}, \bar{\phi}_{t+1}) + 2\zeta\mu^2\mathbb{E}\|\nabla J(w_t)\|^2 + 2\zeta\mu^2\mathbb{E}\{\|\widehat{\nabla J}(w_t) - \nabla J(w_t)\|^2\} \\
&= -\mathbb{E}d^2(\phi_{t+1}, \bar{\phi}_{t+1}) + 2\zeta\mu^2\mathbb{E}\|\nabla J(w_t)\|^2 + 2\zeta\mu^2\mathbb{E}\{\mathbb{E}\{\|\widehat{\nabla J}(w_t) - \nabla J(w_t)\|^2|\mathcal{F}_t\}\} \\
&\le -\mathbb{E}d^2(\phi_{t+1}, \bar{\phi}_{t+1}) + 2\zeta\mu^2\mathbb{E}\|\nabla J(w_t)\|^2 + 2\zeta\mu^2\sigma^2,
\end{aligned} \tag{62}$$

where we use the fact $\frac{1}{2}(a + b)^2 \le a^2 + b^2$ in the second equality, and (18) from Assumption 5.7 in the third inequality. Take expectation of (57) w.r.t. $\{x_s\}_{s=0}^t$ and combine the result with (62), we obtain

$$\mathbb{E}P(\phi_t) \le \frac{1 - \alpha h_{min}}{2\alpha}\mathbb{E}d^2(\phi_t, \bar{\phi}_t) - \frac{1}{2\alpha}\mathbb{E}d^2(\phi_{t+1}, \bar{\phi}_{t+1}) + \frac{\zeta\alpha}{2}\mathbb{E}\|\nabla P(\phi_t)\|^2 + \frac{\zeta\mu^2}{\alpha}\mathbb{E}\|\nabla J(w_t)\|^2 + \frac{\zeta\mu^2\sigma^2}{\alpha}. \tag{63}$$

Multiplying (22) in Lemma 5.11 by $2\zeta$ and summing the result to (63), and considering the upper bound of $\|\nabla J(\boldsymbol{w}_t)\|^2$ given in Lemma 5.10, we have

$$2\zeta\mathbb{E}P(\phi_{t+1}) - (2\zeta - 1)\mathbb{E}P(\phi_t) \le \frac{1 - \alpha h_{min}}{2\alpha}\mathbb{E}d^2(\phi_t, \bar{\phi}_t) - \frac{1}{2\alpha}\mathbb{E}d^2(\phi_{t+1}, \bar{\phi}_{t+1}) + \frac{11\zeta\mu^2}{\alpha}G^2 + \frac{3\zeta\mu^2}{\alpha}\sigma^2. \quad (64)$$

Multiplying (64) by $(1 - \tau)^{-t}$, we have:

$$(1 - \tau)^{-t}2\zeta\mathbb{E}P(\phi_{t+1}) - (1 - \tau)^{-t}(1 - \frac{1}{2\zeta})2\zeta\mathbb{E}P(\phi_t) \le (1 - \tau)^{-t}\frac{1 - \alpha h_{min}}{2\alpha}\mathbb{E}d^2(\phi_t, \bar{\phi}_t)$$
$$- (1 - \tau)^{-t}\frac{1}{2\alpha}\mathbb{E}d^2(\phi_{t+1}, \bar{\phi}_{t+1})$$
$$+ (1 - \tau)^{-t}\frac{11\zeta\mu^2}{\alpha}G^2 + (1 - \tau)^{-t}\frac{3\zeta\mu^2}{\alpha}\sigma^2. \quad (65)$$

Now we sum (65) from $t = 0$ to $t = s - 1$. To simplify the summation, we consider the case $t = 0$ and $t \ge 1$ separately as we can get a simpler upper bound in the latter case. Consider the case $t = 0$, which is simple. From (64) we have:

$$2\zeta\mathbb{E}P(\phi_1) - (2\zeta - 1)\mathbb{E}P(\phi_0) \le \frac{1 - \alpha h_{min}}{2\alpha}\mathbb{E}d^2(\phi_0, \bar{\phi}_0) - \frac{1}{2\alpha}\mathbb{E}d^2(\phi_1, \bar{\phi}_1) + \frac{11\zeta\mu^2}{\alpha}G^2 + \frac{3\zeta\mu^2}{\alpha}\sigma^2. \quad (66)$$

For the case $t \ge 1$, inspired by (Zhang & Sra, 2016), let $\tau = \min\{\frac{1}{2\zeta}, \alpha h_{min}\}$, this implies $\tau \le \frac{1}{2\zeta}$ and $\tau \le \alpha h_{min}$. Consider $\alpha \le h_{max}^{-1} < h_{min}^{-1}$, we have $\tau \in (0, 1)$. For $t \ge 1$, from (65) we can obtain:

$$(1 - \tau)^{-t}2\zeta\mathbb{E}P(\phi_{t+1}) - (1 - \tau)^{-(t-1)}2\zeta\mathbb{E}P(\phi_t) \le (1 - \tau)^{-(t-1)}\frac{1}{2\alpha}\mathbb{E}d^2(\phi_t, \bar{\phi}_t) - (1 - \tau)^{-t}\frac{1}{2\alpha}\mathbb{E}d^2(\phi_{t+1}, \bar{\phi}_{t+1})$$
$$+ (1 - \tau)^{-t}\frac{11\zeta\mu^2}{\alpha}G^2 + (1 - \tau)^{-t}\frac{3\zeta\mu^2}{\alpha}\sigma^2. \quad (67)$$

Finally, summing (65) over $t$ from $t = 0$ to $t = s - 1$, and using the previous results, we have:

$$(1 - \tau)^{-(s-1)}2\zeta\mathbb{E}P(\phi_s) - (2\zeta - 1)\mathbb{E}P(\phi_0) \le \frac{1 - \alpha h_{min}}{2\alpha}\mathbb{E}d^2(\phi_0, \bar{\phi}_0) - (1 - \tau)^{-(s-1)}\frac{1}{2\alpha}\mathbb{E}d^2(\phi_s, \bar{\phi}_s)$$
$$+ \sum_{t=0}^{s-1}(1 - \tau)^{-t}\frac{11\zeta\mu^2}{\alpha}G^2 + \sum_{t=0}^{s-1}(1 - \tau)^{-t}\frac{3\zeta\mu^2}{\alpha}\sigma^2$$
$$\le \frac{D^2}{2\alpha} + \sum_{t=0}^{s-1}(1 - \tau)^{-t}\frac{11\zeta\mu^2}{\alpha}G^2 + \sum_{t=0}^{s-1}(1 - \tau)^{-t}\frac{3\zeta\mu^2}{\alpha}\sigma^2, \quad (68)$$

where the second inequality drops the negative terms and plugs in $d(\phi_0, \bar{\phi}_0) \le D$ (Assumption 5.1).

Define $\gamma_4(\beta) \triangleq \exp_{\phi_0}\left(\beta \exp_{\phi_0}^{-1}(\bar{\phi}_0)\right)$ as the minimal geodesic from $\phi_0$ to $\bar{\phi}_0$. Using the second-order Taylor expansion of $\beta \mapsto P(\gamma_4(\beta))$ around $\beta = 1$, considering $P(\bar{\phi}_0) = 0$, $\nabla P(\bar{\phi}_0) = 0$, we have (Afsari et al., 2013):

$$P(\phi_0) \le P(\bar{\phi}_0) + \langle\nabla P(\bar{\phi}_0), \exp_{\phi_0}^{-1}(\bar{\phi}_0)\rangle + \frac{h_{max}\|\exp_{\phi_0}^{-1}(\bar{\phi}_0)\|^2}{2}$$
$$= \frac{h_{max}}{2}d^2(\phi_0, \bar{\phi}_0). \quad (69)$$

This ensures $P(\phi_0) \le \frac{h_{max}}{2}D^2 \le \frac{D^2}{2\alpha}$ since $d(\phi_0, \bar{\phi}_0) \le D$ and $\alpha \in (0, h_{max}^{-1}]$, one can thus obtain from (68) that

$$\mathbb{E}P(\phi_s) \le \frac{(1 - \tau)^{(s-1)}D^2}{2\alpha} + \sum_{t=0}^{s-1}(1 - \tau)^t\frac{11\mu^2}{2\alpha}G^2 + \sum_{t=0}^{s-1}(1 - \tau)^t\frac{3\mu^2}{2\alpha}\sigma^2$$
$$\le \frac{(1 - \tau)^{(s-1)}D^2}{2\alpha} + \sum_{t=0}^{\infty}(1 - \tau)^t\frac{11\mu^2}{2\alpha}G^2 + \sum_{t=0}^{\infty}(1 - \tau)^t\frac{3\mu^2}{2\alpha}\sigma^2$$

$$\leq \frac{(1-\tau)^{(s-1)}D^2}{2\alpha} + \frac{11\mu^2}{2\alpha\tau}G^2 + \frac{3\mu^2}{2\alpha\tau}\sigma^2$$

$$\leq \frac{11\mu^2}{2\alpha\tau}G^2 + \frac{3\mu^2}{\alpha\tau}\sigma^2 \,, \tag{70}$$

where the last inequality holds whenever:

$$\frac{(1-\tau)^{(s-1)}D^2}{2\alpha} \leq \frac{3\mu^2}{2\alpha\tau}\sigma^2 \iff (1-\tau)^{(s-1)} \leq \frac{3\mu^2}{\tau D^2}\sigma^2$$

$$\iff (s-1)\log(1-\tau) \leq 2\log(\mu) + O(1)$$

$$\iff s \leq \frac{2\log(\mu)}{\log(1-\tau)} + O(1) \,. \tag{71}$$

We conclude that

$$\mathbb{E}\{P(\boldsymbol{\phi}_s)\} \leq \frac{11\mu^2}{2\alpha\tau}G^2 + \frac{3\mu^2}{\alpha\tau}\sigma^2 \,, \tag{72}$$

with sufficiently small step sizes $\mu$ after sufficient iterations $s_o$, where

$$s_o = \frac{2\log(\mu)}{\log(1-\tau)} + O(1) = O(\mu^{-1}) \tag{73}$$

where the second equality follows since $\lim_{\mu\to 0}\mu\log(\mu) = 0$, which means that the magnitude of $\log(\mu)$ can be bounded above by a constant multiple of $\mu^{-1}$ for $\mu \to 0$.

## B.2. Theorem 5.15

Denote $\Delta_t = J(\boldsymbol{w}_t) - J(\boldsymbol{w}^*)$, from Lemma 5.14, we have:

$$\mathbb{E}\Delta_{t+1} - \mathbb{E}\Delta_t \leq -\frac{\mu}{4}\mathbb{E}\|\widehat{\nabla J}(\boldsymbol{w}_t)\|^2 + \frac{5\alpha^2}{\mu}\mathbb{E}\|\nabla P(\boldsymbol{\phi}_{t+1})\|^2 \,. \tag{74}$$

From Corollary 5.4, for the update $\boldsymbol{\phi}_{t+1} = \exp_{\boldsymbol{w}_t}\left(-\mu\widehat{\nabla J}(\boldsymbol{w}_t)\right)$ in (6), we have

$$\langle -\widehat{\nabla J}(\boldsymbol{w}_t), \exp_{\boldsymbol{w}_t}^{-1}(\boldsymbol{w}^*)\rangle \leq \frac{1}{2\mu}d^2(\boldsymbol{w}_t, \boldsymbol{w}^*) - \frac{1}{2\mu}d^2(\boldsymbol{\phi}_{t+1}, \boldsymbol{w}^*) + \frac{\zeta\mu}{2}\|\widehat{\nabla J}(\boldsymbol{w}_t)\|^2 \,. \tag{75}$$

Take the expectation on the previous result w.r.t. $\{\boldsymbol{x}_s\}_{s=0}^t$ and consider (17) in Assumption 5.7, we obtain

$$\mathbb{E}\{\langle -\widehat{\nabla J}(\boldsymbol{w}_t), \exp_{\boldsymbol{w}_t}^{-1}(\boldsymbol{w}^*)\rangle\} = \mathbb{E}\{\langle -\mathbb{E}\{\widehat{\nabla J}(\boldsymbol{w}_t)|\mathcal{F}_t\}, \exp_{\boldsymbol{w}_t}^{-1}(\boldsymbol{w}^*)\rangle\}$$

$$= \mathbb{E}\{\langle -\nabla J(\boldsymbol{w}_t), \exp_{\boldsymbol{w}_t}^{-1}(\boldsymbol{w}^*)\rangle\} \,, \tag{76}$$

Combining (75) and (76), we have

$$\mathbb{E}\{\langle -\nabla J(\boldsymbol{w}_t), \exp_{\boldsymbol{w}_t}^{-1}(\boldsymbol{w}^*)\rangle\} \leq \frac{1}{2\mu}\mathbb{E}d^2(\boldsymbol{w}_t, \boldsymbol{w}^*) - \frac{1}{2\mu}\mathbb{E}d^2(\boldsymbol{\phi}_{t+1}, \boldsymbol{w}^*) + \frac{\zeta\mu}{2}\mathbb{E}\|\widehat{\nabla J}(\boldsymbol{w}_t)\|^2 \,, \tag{77}$$

Consider that $J$ is a geodesically convex function under Assumption 5.5, from (77) one can obtain

$$\mathbb{E}\Delta_t = \mathbb{E}\{J(\boldsymbol{w}_t) - J(\boldsymbol{w}^*)\} \leq \mathbb{E}\{\langle -\nabla J(\boldsymbol{w}_t), \exp_{\boldsymbol{w}_t}^{-1}(\boldsymbol{w}^*)\rangle\}$$

$$\leq \frac{1}{2\mu}\mathbb{E}d^2(\boldsymbol{w}_t, \boldsymbol{w}^*) - \frac{1}{2\mu}\mathbb{E}d^2(\boldsymbol{\phi}_{t+1}, \boldsymbol{w}^*) + \frac{\zeta\mu}{2}\mathbb{E}\|\widehat{\nabla J}(\boldsymbol{w}_t)\|^2 \,. \tag{78}$$

Now we need to upper bound $-d_{\mathcal{M}}^2(\boldsymbol{\phi}_{t+1}, \boldsymbol{w}^*)$. From Corollary 5.4, for the update $\boldsymbol{w}_{t+1} = \exp_{\boldsymbol{\phi}_{t+1}}\left(-\alpha\nabla P(\boldsymbol{\phi}_{t+1})\right)$ in (7), we have

$$d^2(\boldsymbol{w}_{t+1}, \boldsymbol{w}^*) - d^2(\boldsymbol{\phi}_{t+1}, \boldsymbol{w}^*) \leq \zeta\alpha^2\|\nabla P(\boldsymbol{\phi}_{t+1})\|^2 + 2\alpha\langle\nabla P(\boldsymbol{\phi}_{t+1}), \exp_{\boldsymbol{\phi}_{t+1}}^{-1}(\boldsymbol{w}^*)\rangle$$

$$\leq \zeta\alpha^2\|\nabla P(\boldsymbol{\phi}_{t+1})\|^2 + 2\alpha\big(P(\boldsymbol{w}^*) - P(\boldsymbol{\phi}_{t+1})\big)$$
$$\leq \zeta\alpha^2\|\nabla P(\boldsymbol{\phi}_{t+1})\|^2 \,, \tag{79}$$

where the second inequality uses the convexity property of $P$ with (13), and the third inequality uses the fact $P(\boldsymbol{w}^*) = 0$ and $P(\boldsymbol{\phi}_{t+1}) \geq 0$.

Take the expectation on the previous result w.r.t. $\{\boldsymbol{x}_s\}_{s=0}^t$, and combine the result with (78), we have

$$\mathbb{E}\Delta_t \leq \frac{1}{2\mu}\mathbb{E}d^2(\boldsymbol{w}_t, \boldsymbol{w}^*) - \frac{1}{2\mu}\mathbb{E}d^2(\boldsymbol{w}_{t+1}, \boldsymbol{w}^*) + \frac{\zeta\mu}{2}\mathbb{E}\|\widehat{\nabla J}(\boldsymbol{w}_t)\|^2 + \frac{\zeta\alpha^2}{2\mu}\mathbb{E}\|\nabla P(\boldsymbol{\phi}_{t+1})\|^2 \,. \tag{80}$$

Multiplying (74) by $2\zeta$ and adding to (80), we have:

$$2\zeta\mathbb{E}\Delta_{t+1} - (2\zeta - 1)\mathbb{E}\Delta_t \leq \frac{1}{2\mu}\mathbb{E}d^2(\boldsymbol{w}_t, \boldsymbol{w}^*) - \frac{1}{2\mu}\mathbb{E}d^2(\boldsymbol{w}_{t+1}, \boldsymbol{w}^*) + \frac{21\zeta\alpha^2}{2\mu}\mathbb{E}\|\nabla P(\boldsymbol{\phi}_{t+1})\|^2 \,. \tag{81}$$

From Corollary 5.13, we know $\mathbb{E}\|\nabla P(\boldsymbol{\phi}_t)\|^2$ converges to a small value after sufficient iterations $s_o$, now we tend to study the convergence of our algorithm after $s_o$ iterations. For $t \geq s_o$ where $s_o$ defined in (24), from (27) one can rewrite (81) as

$$2\zeta\mathbb{E}\Delta_{t+1} - (2\zeta - 1)\mathbb{E}\Delta_t \leq \frac{1}{2\mu}\mathbb{E}d^2(\boldsymbol{w}_t, \boldsymbol{w}^*) - \frac{1}{2\mu}\mathbb{E}d^2(\boldsymbol{w}_{t+1}, \boldsymbol{w}^*) + \frac{231\zeta\alpha\mu}{2\tau}G^2 + \frac{63\zeta\alpha\mu}{\tau}\sigma^2 \,. \tag{82}$$

Summing (82) from $t = s_o$ to $t = s - 1$ and plugging in $d(\boldsymbol{w}_{s_o}, \boldsymbol{w}^*) \leq D$, we obtain

$$2\zeta\mathbb{E}\Delta_s + \sum_{t=s_o+1}^{s-1}\mathbb{E}\Delta_t \leq (2\zeta - 1)\mathbb{E}\Delta_{s_o} + \frac{1}{2\mu}d^2(\boldsymbol{w}_{s_o}, \boldsymbol{w}^*) - \frac{1}{2\mu}\mathbb{E}d^2(\boldsymbol{w}_s, \boldsymbol{w}^*) + (s - s_o)\left(\frac{231\zeta\alpha\mu}{2\tau}G^2 + \frac{63\zeta\alpha\mu}{\tau}\sigma^2\right)$$
$$\leq (2\zeta - 1)\mathbb{E}\Delta_{s_o} + \frac{1}{2\mu}d^2(\boldsymbol{w}_{s_o}, \boldsymbol{w}^*) + (s - s_o)\left(\frac{231\zeta\alpha\mu}{2\tau}G^2 + \frac{63\zeta\alpha\mu}{\tau}\sigma^2\right)$$
$$\leq (2\zeta - 1)\mathbb{E}\Delta_{s_o} + \frac{D^2}{2\mu} + (s - s_o)\left(\frac{231\zeta\alpha\mu}{2\tau}G^2 + \frac{63\zeta\alpha\mu}{\tau}\sigma^2\right) \,. \tag{83}$$

Recall the geodesic $L$-smoothness of $J$ in Assumption 5.6 and plugging into $d(\boldsymbol{w}_{s_o}, \boldsymbol{w}^*) \leq D$ and $\nabla J(\boldsymbol{w}^*) = 0$, we have:

$$\Delta_{s_o} = J(\boldsymbol{w}_{s_o}) - J(\boldsymbol{w}^*) \leq \langle\nabla J(\boldsymbol{w}^*), \exp_{\boldsymbol{w}^*}^{-1}(\boldsymbol{w}_{s_o})\rangle + \frac{L}{2}\|\exp_{\boldsymbol{w}^*}^{-1}(\boldsymbol{w}_{s_o})\|^2$$
$$= \frac{L}{2}d^2(\boldsymbol{w}_{s_o}, \boldsymbol{w}^*) \leq \frac{LD^2}{2} \,. \tag{84}$$

This ensures $\Delta_{s_o} \leq \frac{LD^2}{2} \leq \frac{D^2}{2\mu}$ since $\mu \leq L^{-1}$, so that from (83) one can obtain

$$2\zeta\mathbb{E}\Delta_s + \sum_{t=s_o+1}^{s-1}\mathbb{E}\Delta_t \leq \zeta LD^2 + (s - s_o)\left(\frac{231\zeta\alpha\mu}{2\tau}G^2 + \frac{63\zeta\alpha\mu}{\tau}\sigma^2\right) \,. \tag{85}$$

Here the term $\Delta_s$ does not cancel nicely due to the presence of the curvature term $\zeta$, which necessities the use of a Lyapunov function as in (Zhang & Sra, 2016). Introduce auxiliary variables $\boldsymbol{w}'_{s_o+1} = \boldsymbol{w}_{s_o+1}$ and $\boldsymbol{w}'_{t+1} = \exp_{\boldsymbol{w}'_t}\left(\frac{1}{t-s_o+1}\exp_{\boldsymbol{w}'_t}^{-1}(\boldsymbol{w}_{t+1})\right)$ for $s_o + 1 \leq t \leq s - 2$, $\boldsymbol{w}'_{s+1} = \exp_{\boldsymbol{w}'_{s-1}}\left(\frac{2\zeta}{2\zeta+s-s_o-1}\exp_{\boldsymbol{w}'_{s-1}}^{-1}(\boldsymbol{w}_s)\right)$, repeatedly consider (12) in Assumption 5.5 (geodesic convexity of $J$), for $s \geq s_o + 1$, we have

$$J(\boldsymbol{w}'_{s-1}) \leq \frac{s - s_o - 2}{s - s_o - 1}J(\boldsymbol{w}'_{s-2}) + \frac{1}{s - s_o - 1}J(\boldsymbol{w}_{s-1})$$
$$\leq \frac{s - s_o - 2}{s - s_o - 1}\left(\frac{s - s_o - 3}{s - s_o - 2}J(\boldsymbol{w}'_{s-3}) + \frac{1}{s - s_o - 2}J(\boldsymbol{w}_{s-2})\right) + \frac{1}{s - s_o - 1}J(\boldsymbol{w}_{s-1})$$
$$\leq \cdots \leq \frac{1}{s - s_o - 1}\sum_{t=s_o+1}^{s-1}J(\boldsymbol{w}_t) \,. \tag{86}$$

Denote $\Delta'_s = J(\boldsymbol{w}'_s) - J(\boldsymbol{w}^*)$, we have $\mathbb{E}\Delta'_{s-1} \leq \frac{1}{s-s_o-1}\sum_{t=s_o+1}^{s-1}\mathbb{E}\Delta_t$. Again, consider the geodesic convexity of $J$ in (12) of Assumption 5.5, and we can further write

$$\mathbb{E}\Delta'_s = \mathbb{E}\{J(\boldsymbol{w}'_s) - J(\boldsymbol{w}^*)\} \leq \mathbb{E}\left\{\frac{s-s_o-1}{2\zeta+s-s_o-1}J(\boldsymbol{w}'_{s-1}) + \frac{2\zeta}{2\zeta+s-s_o-1}J(\boldsymbol{w}_s) - J(\boldsymbol{w}^*)\right\}$$

$$= \frac{2\zeta\mathbb{E}\Delta_s + (s-s_o-1)\mathbb{E}\Delta'_{s-1}}{2\zeta+s-s_o-1}$$

$$\leq \frac{2\zeta\mathbb{E}\Delta_s + \sum_{t=s_o+1}^{s-1}\mathbb{E}\Delta_t}{2\zeta+s-s_o-1}. \tag{87}$$

Plug the upper bound of $2\zeta\mathbb{E}\Delta_s + \sum_{t=s_o+1}^{s-1}\mathbb{E}\Delta_t$ in (83) into the above result, for $s \geq s_o+1$, we have

$$\mathbb{E}\Delta'_s \leq \frac{\zeta L D^2}{2\zeta+s-s_o-1} + \frac{s-s_o}{2\zeta+s-s_o-1}\left(\frac{231\zeta\alpha\mu}{2\tau}G^2 + \frac{63\zeta\alpha\mu}{\tau}\sigma^2\right). \tag{88}$$

## C. Examples of Riemannian manifolds

### C.1. Grassmann manifold

The Grassmann manifold $\mathcal{G}_n^p$, a set of $p$-dimensional linear subspaces of $\mathbb{R}^n$, can be regarded as a smooth quotient manifold of the Stiefel manifold $\mathcal{S}_n^p = \{\boldsymbol{U} \in \mathbb{R}^{n\times p} : \boldsymbol{U}^T\boldsymbol{U} = \boldsymbol{I}_p\}$, i.e., $\mathcal{G}_n^p = \mathcal{S}_n^p/\mathcal{O}_p = \{\pi(\boldsymbol{U}) : \boldsymbol{U} \in \mathcal{S}_n^p\}$ where $\mathcal{O}_p = \{\boldsymbol{U} \in \mathbb{R}^{p\times p} : \boldsymbol{U}^T\boldsymbol{U} = \boldsymbol{I}_p\}$ is the orthogonal group and $\pi : \mathcal{S}_n^p \to \mathcal{G}_n^p$ is the map $\pi(\boldsymbol{U}) = \{\boldsymbol{UO} : \boldsymbol{O} \in \mathcal{O}_p\}$. The geodesic distance between two subspaces $\pi(\boldsymbol{U}_1)$ and $\pi(\boldsymbol{U}_2)$ of $\mathcal{G}_n^p$, spanned by orthonormal matrices $\boldsymbol{U}_1$ and $\boldsymbol{U}_2$, is defined as follows (Edelman et al., 1998):

$$d_{\mathcal{G}_n^p}(\boldsymbol{U}_1, \boldsymbol{U}_2) = \|\cos^{-1}(\boldsymbol{\theta})\|_2, \tag{89}$$

where $\boldsymbol{\theta} \in \mathbb{R}^p$ contains the singular values of $\boldsymbol{U}_1^T\boldsymbol{U}_2$, namely, it is related to its singular value decomposition (SVD) as $\boldsymbol{U}_1^T\boldsymbol{U}_2 = \boldsymbol{V}_1^T\mathrm{diag}(\boldsymbol{\theta})\boldsymbol{V}_2$. Define $\bar{f} : \mathcal{S}_n^p \to \mathbb{R}$, we have $f(\pi(\boldsymbol{U})) = \bar{f}(\boldsymbol{U})$ for all $\pi(\boldsymbol{U}) \in \mathcal{G}_n^p$. The Riemannian gradient $\nabla f$ at $\pi(\boldsymbol{U}) \in \mathcal{G}_n^p$ is given by:

$$\nabla f(\pi(\boldsymbol{U})) = \nabla\bar{f}(\boldsymbol{U}) = \boldsymbol{P}_{\boldsymbol{U}}^{\mathcal{G}_n^p}(\boldsymbol{G}), \tag{90}$$

with $\boldsymbol{P}_{\boldsymbol{U}}^{\mathcal{G}_n^p}(\boldsymbol{G}) = (\boldsymbol{I} - \boldsymbol{UU}^T)\boldsymbol{G}$, where $\boldsymbol{G} \in \mathbb{R}^{n\times p}$ is the Euclidean gradient of $\bar{f}$ at $\boldsymbol{U}$. Let $\boldsymbol{\xi} \in T_{\pi(\boldsymbol{U})}\mathcal{G}_n^p$, and let $\boldsymbol{X\Sigma Y} = \boldsymbol{U} + \boldsymbol{\xi}$ be the thin SVD of $\boldsymbol{U} + \boldsymbol{\xi} \in \mathbb{R}^{n\times p}$. A numerically stable retraction $R_{\pi(\boldsymbol{U})} : T_{\pi(\boldsymbol{U})}\mathcal{G}_n^p \to \mathcal{G}_n^p$ on $\mathcal{G}_n^p$ is given by (Boumal, 2023):

$$R_{\pi(\boldsymbol{U})}(\boldsymbol{\xi}) = \pi(\boldsymbol{XY}^T). \tag{91}$$

### C.2. The manifold of SPD matrices

The geodesic distance of $\mathcal{S}_n^{++}$ between two SPD matrices $\boldsymbol{\Sigma}_1$ and $\boldsymbol{\Sigma}_2 \in \mathcal{S}_n^{++}$ can be computed in closed form (Pennec et al., 2006) as:

$$d_{\mathcal{S}_n^{++}}(\boldsymbol{\Sigma}_1, \boldsymbol{\Sigma}_2) = \left\|\log(\boldsymbol{\Sigma}_2^{-\frac{1}{2}}\boldsymbol{\Sigma}_1\boldsymbol{\Sigma}_2^{-\frac{1}{2}})\right\|_F, \tag{92}$$

where $\|\cdot\|_F$ denotes the Frobenius norm. The Riemannian gradient $\nabla f$ at $\boldsymbol{\Sigma} \in \mathcal{S}_n^{++}$ is given by:

$$\nabla f(\boldsymbol{\Sigma}) = \boldsymbol{\Sigma}\,\mathrm{sym}(\boldsymbol{G})\boldsymbol{\Sigma}, \tag{93}$$

with $\boldsymbol{G} \in \mathbb{R}^{p\times p}$ the Euclidean gradient of function $f$ at $\boldsymbol{\Sigma}$ and $\mathrm{sym}(\boldsymbol{G}) = \frac{1}{2}(\boldsymbol{G}^T + \boldsymbol{G})$. In practice, the Euclidean gradient can be easily computed using automatic differentiation tools. Let $\boldsymbol{\xi} \in T_{\boldsymbol{\Sigma}}\mathcal{S}_n^{++}$. A retraction $R_{\boldsymbol{\Sigma},\mathcal{S}_n^{++}} : T_{\boldsymbol{\Sigma}}\mathcal{S}_n^{++} \mapsto \mathcal{S}_n^{++}$ is defined as:

$$R_{\boldsymbol{\Sigma},\mathcal{S}_n^{++}}(\boldsymbol{\xi}) = \boldsymbol{\Sigma} + \boldsymbol{\xi} + \frac{1}{2}\boldsymbol{\xi}\boldsymbol{\Sigma}^{-1}\boldsymbol{\xi}. \tag{94}$$

This retraction is a second-order approximation of the exponential mapping.

# D. Additional experimental results

### D.1. Inefficiency of the method (Wang et al., 2024b)

In Section 2, we argue that the algorithm in (Wang et al., 2024b) is inefficient due to the inner-loop optimization when minimizing the penalty term $P(\phi_t)$. To support this claim, we compare the MSD performance and runtime between the work in (Wang et al., 2024b) (denoted as "Inefficient Riemannian diffusion") and the proposed algorithm. We examine these two algorithms for distributed PCA on synthetic data in the same setting as in Subsection 7.1, and produce the results as in Figure 6. From these results, we can see that while the performance of these two algorithms is nearly identical, the proposed algorithm achieves a significantly reduced runtime. These experiments were performed on a computer with an Apple M4 Pro processor and 24GB of RAM.

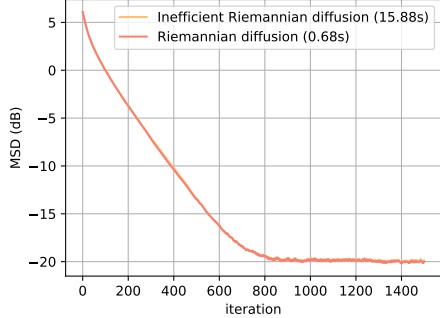

*Figure 6.* Illustration of MSD performance and runtime per Monte Carlo run of the (inefficient) Riemannian diffusion adaptation algorithms for distributed PCA on synthetic data.

### D.2. Applicability to more networks

To illustrate the applicability of the proposed algorithm to more networks, we randomly generate another graph topology as shown in Figure 7 (left) and select weights with an uniform rule. We test all compared algorithms for both distributed PCA and GMM inference on synthetic data in the same setting as in Section 7 and produce experimental results in Figure 7 (middle and right). From these results, we find the performance of the compared algorithms remains similar to that shown in Figure 2 and Figure 4, which are obtained with the network illustrated in Figure 1.

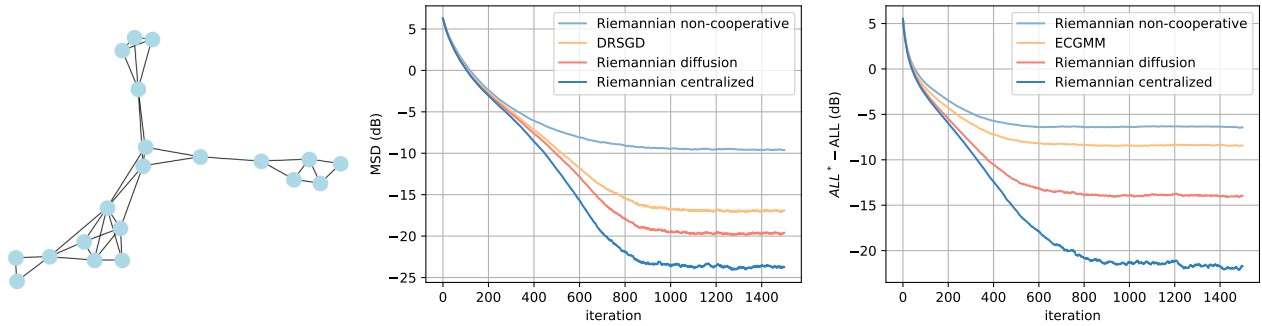

*Figure 7.* Performance illustration of the compared methods on another network with uniformly distributed weights: graph topology (left); MSD performance for distributed PCA (middle); ALL differences for GMM inference (right) on synthetic data.

### D.3. Impact of step sizes

For the proposed algorithm, the choice of step sizes is critical to control the tradeoff between convergence speed and steady-state performance. We examine the behavior of the proposed algorithm for distributed PCA on synthetic data in the same setting as in Subsection 7.1, and produce the results with different choices of step sizes as shown in Figure 8. It can be observed that larger step sizes tend to accelerate convergence but result in worse performance at steady state.

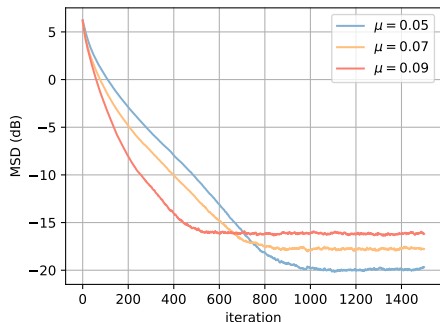

*Figure 8.* Illustration of MSD performance of the proposed method with different step sizes for distributed PCA on synthetic data.

## E. Computational complexity

The computational complexity of the proposed algorithm on each agent $k$ involves two contributing terms. The first is the cost of a local adaptation step (3) (i.e., Riemannian SGD on $J_k$), which is denoted by $\mathsf{T}_J$. The second is the cost of the combination step (4), which involves a gradient step over the loss function $P_k$ that scales linearly with the number $N_{\mathrm{neigh},k}$ of neighbors connected to node $k$ in the graph (that is, with the number of nonzero elements in the coefficients $c_{k\ell}$), which we represent as $N_{\mathrm{neigh},k} \cdot \mathsf{T}_P$, where $\mathsf{T}_P$ is the cost of computing the inverse of the exponential mapping. $N_{\mathrm{neigh},k}$ is also known as the *degree* of the vertex $k$ in the graph $\mathcal{G}$. Thus, for each agent $k$, we obtain a complexity of $\mathsf{T}_J + N_{\mathrm{neigh},k} \cdot \mathsf{T}_P$. Compared to a non-cooperative setting, we have an overhead cost of $N_{\mathrm{neigh},k} \cdot \mathsf{T}_P$, which is a function of both on $\mathsf{T}_P$ (which depends on the manifold) and on the number of neighbors connected to node $k$ (which depends on the graph topology).

This allows us to understand how the complexity scales with the number of agents $K$. In the case where the number of neighbors to each node (i.e., their degree in the graph) is constant, the complexity does not increase with $K$. On the other hand, in the worst case scenario of a fully connected graph (where each vertex has degree $K-1$, being connected to all other vertices), then the complexity scales linearly with $K$, with a coefficient equal to $\mathsf{T}_P$.

## F. Discussion on the limitations

Our work has two main limitations, which are discussed in the following.

- The theoretical analysis is based on the exponential mapping $\exp_x$ as in many works in Riemannian optimization, e.g., (Zhang & Sra, 2016), while in practice, a retraction $R_x$ is used for more efficient computations. A key result in (Bonnabel, 2013) states that $d(R_x(\mu \cdot v), \exp_x(\mu \cdot v)) = O(\mu^2)$, meaning that for small $\mu$, a retraction closely approximates the exponential map. The main approach to proving convergence with retractions involves showing that the iterates of the algorithm remain close to those of an equivalent version using the exponential map, which holds as $\mu \to 0$ (Bonnabel, 2013). This argument typically relies on diminishing step sizes, whereas our analysis is designed for constant step sizes, which are crucial for continuous adaptation and learning. Some works also employ the *pullback* operator $f \circ R_x$, i.e., the composition of the cost function $f$ and a retraction $R_x$, to establish convergence. However, these approaches require assumptions that may be less natural, such as the convexity and smoothness of the pullback operator, see Chapter 4 of (Boumal, 2023). Thus, we believe that extending the proposed theoretical analysis based on a retraction is an exciting, though non-trivial, research direction.

- Manifolds without closed-form expressions for retractions, or for the Riemannian gradient, pose challenges to the implementation of the proposed algorithm, as such operations have to be approximated numerically in some way. However, we highlight that this limitation also applies to most existing Riemannian optimization algorithms and is not specific to our work.

