# OpenReview forum: "Riemannian Diffusion Adaptation for Distributed Optimization on Manifolds"
_ICML.cc/2025/Conference — ICML 2025 poster_

### Official Review · Reviewer_tsm9 · 2025-03-11

**Overall Recommendation:** 3

**Summary:**

The paper concerns online distributed optimization for data on Riemannian manifolds. The authors propose an algorithm for distributed optimization between a number of agents with combination steps in the tangent spaces of the current values at the agents. The paper contains a theoretical analysis of the algorithm, and experimental validation on synthetic and real data.

## update after rebuttal
I have maintained my initial score as it still adequately reflects my evaluation of the paper.

**Claims And Evidence:**

yes

**Essential References Not Discussed:**

no

**Experimental Designs Or Analyses:**

I did not find any issues

**Methods And Evaluation Criteria:**

yes

**Other Comments Or Suggestions:**

no other comment

**Other Strengths And Weaknesses:**

Strengths:
- well-written paper. Clear explanation of the chosen methodology
- well-chosen methodology to solve the problem
- thorough theoretical analysis of the proposed algorithm

Weaknesses:
- I believe the algorithm is a fairly straight-forward generalization of its Euclidean counterpart. This is not necessarily a bad thing, it just mean that the algorithm itself does contribute significant new ideas. This is partly balanced by the theoretical analysis that has to account for the geometry

**Questions For Authors:**

no questions

**Relation To Broader Scientific Literature:**

while I am not specifically aware of the literature on e.g. decentralized optimization, I believe the literature is adequately surveyed

**Theoretical Claims:**

I did not check the proofs for correctness. My overall impression from reading the main paper is that the exposition is correct

---

> ### Author Rebuttal · Authors · 2025-03-29
>
> Thank you for reviewing our work and pointing out the strengths and weaknesses. In the following, we provide replies to weaknesses:
>
> We would like to emphasize that the proposed work is not a trivial generalization, even though there is an Euclidean counterpart to the Riemannian diffusion adaptation algorithm. For example, when combining the local estimates lying on manifolds, it is no longer possible to use a simple linear combination of the local estimates as in the Euclidean counterpart, as we do not assume a vector space structure in the manifold. We tackled this issue by proposing a one-step Riemannian gradient descent over a network agreement loss function to achieve information exchange during the learning and adaptation process.
>
> As you also mentioned, the theoretical analysis (both in terms of the network agreement and non-asymptotic convergence) involves fundamentally new ideas since our analysis is fully geometric. For example, in the network agreement analysis, we cannot use adjacency matrix decomposition as in the Euclidean counterpart due to the combination step in our case being *non-linear*.  Thus, the analysis is rendered significantly more difficult since the network centroid cannot be computed using a simple/linear expression as in the Euclidean case. Therefore, we propose a novel framework to study the network agreement through the evolution of the penalty term $P(\boldsymbol{\phi}_t)$. In the non-asymptotic convergence analysis, curvature-related terms also make traditional techniques used in Euclidean spaces unfeasible, requiring careful design of a Lyapunov function (please see more details in *[the response to reviewer bCcG](https://openreview.net/forum?id=5tyvHfhRFZ&noteId=IqmTIhR1dX)*).

---

### Official Review · Reviewer_8uHb · 2025-03-12

**Overall Recommendation:** 3

**Summary:**

This paper aims to solve the online decentralized optimization problem on the general Riemannian manifold for multi-agents. The proposed Riemannian diffusion adaptation method contains two stages: an adaptation step and a combination step. It theoretically proves that all agents will approximately converge to a network agreement with non-asymptotic convergence after sufficient iterations. The experiments on two typical manifolds for PCA and GMM show that the proposed method significantly outperforms the non-cooperative, DRSGD and ECGMM methods.

**Claims And Evidence:**

Yes. The authors provide detailed and complete proofs of the theorems.

**Essential References Not Discussed:**

No.

**Experimental Designs Or Analyses:**

If for an intuitive demonstration, the example in distributed PCA and GMM inference on synthetic and real data is great. But for wide application aspects, the experiments need to conduct more complex situations in real word benchmarks.

**Methods And Evaluation Criteria:**

The proposed method is evaluated on synthetic data and real data in application of distributed PCA and GMM inference. However, more complex situations and applications are lacking discussion.

**Other Comments Or Suggestions:**

No other comments.

**Other Strengths And Weaknesses:**

Pros:
1. The proposed methods are concise and clear. The two stages: for the adaptation step, the idea is to compute a local solution for each agent. In the combination step, the idea is making agreements with all agents.

2. The whole idea is very simple, and the theoretical analysis is rich.

3. Compared to previous works, the proposed method avoids transforming points to Euclidean space and back to manifold repeatedly.

Cons:
1. The two examples and applications are all simple cases, it is hard for readers to accept that the proposed method will have wide application.

2. Here are 4 assumptions in this paper, I think it lacks claims that these assumptions will fit most common situations in real applications?

3. The computational complexity of the proposed method is not discussed, in the high-dimensional case, will the computational cost be very high?

**Questions For Authors:**

1. In experimental results, why do the Riemannian centralized methods outperform the proposed methods so much? What are the advantages of the proposed method compared to Riemannian centralized method?
2. The authors said that the proposed method is a strategy over general manifolds. However, in the general manifold case, e.g., the manifold embedded in $\mathbb{R}^{m+n}$ defined by implicit function $p=(u, v) \in \mathbb{R}^{m+n}$ where $v=f(u)$ with $f:\mathbb{R}^m\rightarrow \mathbb{R}^n$, how does the proposed Riemannian diffusion Adaptation process?

**Relation To Broader Scientific Literature:**

The previous works on decentralized optimization are in Euclidean space. When extending to Riemannian manifolds, some previous works construct functions that map the points from the manifold to Euclidean space. But this paper proposes a more direct method, avoiding transformation between manifold and Euclidean space, and it can be proved theoretically that the algorithm will be non-asymptotic convergence.

**Theoretical Claims:**

Yes, I have checked most proofs for lemmas and theories, especially for Lemma 5.11 and Theorem 5.15.

---

> ### Author Rebuttal · Authors · 2025-03-31
>
> Thank you for reading our work (especially in checking the proofs) and offering constructive suggestions. In the following, we provide clarifications and answers to your comments and questions.
>
> ### **Replies to weakness:**
>
> ***Conduct more experiments for wider application aspects:***
>
> Thank you for this comment. We highlight that while PCA and GMM inference are key problems in machine learning, the proposed method is fairly general and applicable to a wide set of problems (the revised manuscript will be modified to better showcase this). Moreover, one of the main contributions of our paper is the theoretical analysis, and we find that the experimental validation in our paper is still in line with those performed in related work (see, e.g., Bonnabel, 2013; Zhang & Sra, 2016; Chen et al., 2021; Vlaski & Sayed, 2021; Li & Ma, 2023). While additional experiments with complex real-world benchmarks would definitely improve the paper, unfortunately, we were not able to perform such experiments in the comparatively short time available during the rebuttal period.
>
> ***Claim the fitness of the assumptions in real applications:***
>
> Thank you for this insightful comment. These main assumptions made in the paper are indeed standard in Riemannian and decentralized optimization algorithms, particularly in what concerns their theoretical analysis, see e.g.,  (Bonnabel,
> 2013; Zhang et al., 2016; Tripuraneni et al., 2018; Afsari, 2011; Chen & Sayed, 2012; Sayed et al., 2013; Afsari et al., 2013). A necessary assumption for the derivation of the algorithm is the smoothness of the risk functions, since the proposed algorithm relies on gradients, while the remaining assumptions are used in the theoretical study. Moreover, an assumption that is less frequently satisfied is geodesic convexity.
> Considering the practical examples studied in the paper (PCA and GMM inference), the cost function of PCA formulated on the Grassmannian manifold has been recently shown to be geodesically convex [R1]. On the other hand, for GMM inference, the log-likelihood has been shown to be geodesically convex for the case of a single Gaussian (Hosseini & Sra, 2015), but not necessarily when multiple Gaussians are considered. From the example of GMM inference, we find the proposed algorithm itself can work even in some situations when not all these assumptions are satisfied.
> We will update the revised manuscript to discuss the applicability of the assumptions in our work to practical problems.
>
> [R1] Alimisis, F., and Vandereycken, B. Geodesic convexity of the symmetric eigenvalue problem and convergence of steepest descent. Journal of Optimization Theory and Applications, 203(1), 920-959, 2024.
>
> ***Discussion for the computational complexity:***
>
> Thank you for this useful comment. While the computation complexity will scale with the dimension of the manifold, the proposed approach is a first-order gradient-based method, and its complexity remains as low as in other first-order optimization approaches such as (Bonnabel, 2013; Zhang & Sra, 2016; Tripuraneni et al., 2018). For a detailed discussion on the computational complexity of the proposed algorithm, please refer to *[the response to Reviewer xKFw](https://openreview.net/forum?id=5tyvHfhRFZ&noteId=Lgg3fpLQRg)*.
>
> ### **Replies to questions:**
>
> ***Why does the Riemannian centralized method perform so much better? What are the advantages of the proposed method?***
>
> The centralized solution achieved the best performance, as it can access all the data over the whole network at every iteration.
> The proposed algorithm is fully decentralized, where each agent uses only locally observed data to update its local estimate and exchange information only among neighboring agents. Although the proposed algorithm has lower performance compared to the centralized method, it can be computed in parallel on multiple agents. We will add more details in the revised manuscript to better explain the differences and advantages.
>
> ***How does the proposed method work on a manifold defined through an implicit function?***
>
> The proposed algorithm is designed for general Riemannian manifolds, but it requires the computation of retractions and Riemannian gradients, as in other Riemannian optimization works (see, e.g., (Boumal, 2023; Zhang & Sra, 2016; Bonnabel, 2013), to name a few). When mentioning "general manifolds", we aimed to differentiate our approach from extrinsic works that focus on specific examples, such as Stiefel manifolds (Chen et al., 2021; Wang & Liu, 2022). For a manifold defined through an implicit function, such a Riemannian structure and the required operations would have to be derived to make those strategies applicable. We will clarify the "general manifolds" statement in the revised manuscript.

---

### Official Review · Reviewer_xKFw · 2025-03-13

**Overall Recommendation:** 4

**Summary:**

This paper proposes a decentralized optimization algorithm on manifolds that is termed Riemannian diffusion adaptation algorithm. The proposed algorithm follows two steps. First, in the adaptation step, each agent updates its local solution estimate on the manifold using Riemannian stochastic gradient descent (R-SGD). Second, in the combination step, agents share and combine their estimates in the tangent space. A theoretical analysis under a constant step size shows that the algorithm achieves network agreement with high probability and converges to a neighborhood of the optimal solution. The method is demonstrated on online decentralized PCA and GMM inference, with experiments on both synthetic and real-world data showing its effectiveness.

**Claims And Evidence:**

The claims are adequately supported by theoretical and experimental results.

**Essential References Not Discussed:**

The references are adequate.

**Experimental Designs Or Analyses:**

The experimental design and analyses are sound and valid.

**Methods And Evaluation Criteria:**

The methods and evaluation criteria are adequate.

**Other Comments Or Suggestions:**

- Page 1, right column, lines 011-025: The discussion is vague and should be more concrete. Many terms are mentioned (e.g., embedding and Whitney embedding) without any introduction. To clarify and strengthen the motivation, this paragraph should be re-written.

- Page 1, right column, description of contributions 1: the paragraph contains many terms whose meaning is not completely clear at this stage (adaptation strategy, fully intrinsic, general manifolds, a sequence of efficient adaptation and combination steps).

- Page 1, right column, description of contributions 2: same as in contribution 1, the paragraph contains many unclear terms (network agreement, decreasing geodesic distance, curvature-dependent, non-asymptotic convergence, proper design of Lyapunov function.

- Sec. 5 contains many existing results (up to Cor. 5.5) - consider separating the old results from the new results.

- In Sec. 6.1 - consider putting the definition of MSD in a non-inline equation for better emphasis.

**Other Strengths And Weaknesses:**

Strengths:
- The addressed problem is important and central.
- To the best of my knowledge, although simple and straightforward, the proposed algorithm is new
- The theoretical analysis presents important properties of the algorithm
- The experiments nicely demonstrate the benefits of the algorithm compared to baselines in two classical applications.

Weaknesses:
- The introduction could be improved (see below)
- A discussion on the computational complexity of the algorithm is missing.
- A discussion on the limitations is missing. For example, how well does the algorithm scale with the number of agents? How is the exp map (or retraction) computed on manifolds without a closed-form expression?

**Questions For Authors:**

- In Sec. 7 - only one network is considered. Why only one? How does the algorithm scale with the number of agents K?

**Relation To Broader Scientific Literature:**

The paper addresses an important and central problem.

**Theoretical Claims:**

I checked the derivations and proofs, and they seem correct.

---

> ### Author Rebuttal · Authors · 2025-03-31
>
> Thank you for reading our work (especially in checking the proofs) and offering constructive suggestions. In the following, we provide clarifications and answers to your comments and questions.
>
> ### **Replies to weakness:**
>
> ***Improve the introduction and technical section:***
>
> Thank you very much for the suggested improvements to the introduction and technical sections. We will provide sharper definitions on Page 1 and add subsections in Section 5 to clearly separate existing results from previous works and our technical results in the revised manuscript.
>
> ***Discussion for the computational complexity:***
>
> Thank you for this insightful comment. The computational complexity of the proposed algorithm involves two contributing terms. The first is the cost of a local adaptation step at each agent $k$ (i.e., Riemannian SGD on $J_k$), which is denoted by $T_J$. The second is the cost of the combination step, which involves a gradient step over the loss function $P_k$ that scales linearly with the number $N_{{\rm neigh},k}$ of neighbors connected to node $k$ in the graph (that is, with the number of nonzero elements in the coefficients $c_{k\ell}$), which we represent as $N_{{\rm neigh},k}\cdot T_{P}$, where $T_P$ is the cost of computing the Riemannian logarithm operator. $N_{{\rm neigh},k}$ is also known as the *degree* of the vertex $k$ in the graph $\mathcal{G}$.
> Thus, for each agent $k$, we obtain a complexity of $T_{J} + N_{{\rm neigh},k}\cdot T_{P}$.
> Compared to a noncooperative setting, we have an overhead cost of $N_{{\rm neigh},k}\cdot T_{P}$, which is a function of both on $T_P$ (depending on the manifold) and on the number of neighbors connected to node $k$ (which depends on the network topology).
>
> This allows us to understand how the complexity scales with the number of agents $K$. In the case where the number of neighbors to each node (i.e., their degree in the graph) is constant, the complexity does not increase with the number of agents. On the other hand, in the worst case scenario of a fully connected network (where each vertex has degree $K-1$, being connected to all other vertices), the complexity scales linearly with $K$, with a coefficient equal to $T_P$.
>
> We will include this discussion on the computational complexity of the algorithm in the revised manuscript. In addition, we will calculate the complexity values of $T_J$ and $T_P$ (in terms of the required number of operations) for the PCA and GMM problems discussed in Section 6 of the paper and include them in the revised manuscript.
>
> ***Discussion for the limitations:***
>
> Thank you for this insightful suggestion. We summarize such a discussion in the following, and will include it in the revised manuscript:
>
> - Scaling with the number of agents: the discussion on the computational complexity (explained in more detail just above) shows how the complexity scales with the number of agents $K$. In particular, the computation complexity of the combination step of the algorithm scales according to the *degree* (number of neighbors) of the vertices of the graph. In the worst case of a fully connected network, this contribution scales linearly with the number of agents $K$.
>
> - Manifolds without closed form expressions: manifolds without closed form expressions for retractions, or for the Riemannian gradient, pose challenges to the implementation of the proposed algorithm, as such operations have to be approximated numerically in some way. However, we highlight that this limitation also holds for most existing Riemannian optimization algorithms, and is not specific to our work.
>
> - Theoretical analysis: one limitation of the theoretical analysis is that it relies on the use of the exponential mapping $\exp_x$, while in practice, the use of a retraction $R_x$, which is more computationally efficient. For more details, please see *[the response to Reviewer qfKq](https://openreview.net/forum?id=5tyvHfhRFZ&noteId=wtPNBvN9LG)*.
>
> ### **Replies to questions:**
>
> ***Why is only one network considered?***
>
> To illustrate the applicability to more networks, we randomly generate a different graph topology with uniformly distributed weights, and test all algorithms in the same setting as in Section 7 of the manuscript. The graph topology and experimental results can be seen in https://ibb.co/cKdYKdZ6, and remain similar to those obtained with the original network. We will include more experiments in the revised manuscript.
>
> ***How does the algorithm scale with the number of agents $K$?***
>
> The computational complexity is related to $K$. The proposed algorithm is parallelizable, with an adaptation cost that is constant per agent and an overhead cost of the combination step. The latter depends on the number of neighbors connected to each node in the graph, and in the worst-case scenario, it can increase linearly with $K$. For more details, please see the discussion on the computational complexity in the previous response.

---

### Official Review · Reviewer_bCcG · 2025-03-17

**Overall Recommendation:** 2

**Summary:**

The paper studies online distributed optimization on manifolds, and proposes Riemannian diffusion adaptation in which each agent keeps running two steps until convergence: 1) execute R-SGD; 2) combine outputs of neighboring agents by running one step of RGD over the associated penalty function which characterizes the network agreement. The proposed algorithm is shown to converge to a consensus with high probability provided that a sufficiently small step-size is used. Similar results are established on the convergence of objective function. Experiments are conducted on two instances, i.e., distributed PCA and distributed GMM inference, showing better performance of the proposed algorithm compared to baselines.

**Claims And Evidence:**

The paper seems to extend the Riemannian diffusion adaptation algorithm proposed in Wang et al., 2024b. The difference is that the minimization of the penalty function there is replaced by a gradient descent step here. But it remains unknown why this replacement performs better. Although the minimization of the penalty is time-consuming, it is possible to require a lot less iterations than the version here with one-gradient step.

The main contribution seems to be theoretical analysis of this algorithm. It could be compared with the results in the Euclidean setting.

Also, it was mentioned that a Lyapunov function is designed. But I didn't see it in the main text.

The step size seems critical to the tradeoff between convergence and accuracy. It would be better if experimental results on it can be reported.

Other minor issues: 1) denominators in Lines 275 and 282 are 0 when s=s0+1

**Essential References Not Discussed:**

No

**Experimental Designs Or Analyses:**

Experimental designs follows Wang at el. 2024.

**Methods And Evaluation Criteria:**

see above

**Other Comments Or Suggestions:**

see above

**Other Strengths And Weaknesses:**

see above

**Questions For Authors:**

see above

**Relation To Broader Scientific Literature:**

This paper provides a theoretical analysis of a modified Riemannian diffusion adaptation algorithm.

**Theoretical Claims:**

In the proof of Lemma 5.1, it said the inequality in Eq. (33) holds by Cauchy-Schwarz inequality. This seems wrong.

---

> ### Author Rebuttal · Authors · 2025-03-31
>
> Thank you for reading our work and offering constructive suggestions. We provide clarifications to your comments as below.
>
> ***Lack of evidence that the replacement performs better:***
>
> Thank you for this comment. We argue that the algorithm in (Wang et al., 2024b) is inefficient due to the inner-loop optimization when minimizing the penalty $P(\boldsymbol{\phi}_t)$. We support this claim with a numerical evaluation reported in *[the response to Reviewer qfKq](https://openreview.net/forum?id=5tyvHfhRFZ&noteId=wtPNBvN9LG)*. From this result, we see that the penality minimization in (Wang et al., 2024b) does not require less iterations than our proposed approach, they actually require nearly identical numbers of iterations to achieve convergence. We also present the result in https://ibb.co/nsDtry65 for your convenience.
>
> ***Comparison of the theoretical results with the Euclidean counterpart:***
>
> Thank you for this insightful comment. Compared to the Euclidean counterpart (Chen & Sayed, 2012; Sayed et al., 2013; Vlaski & Sayed, 2021), one essential difference of our results is the impact of the manifold curvature $\kappa$ (captured in the parameter $\zeta$). For example, our convergence rates can be slower for some highly curved manifolds with $\kappa<0$. Besides, for the network agreement, our results focus on the evolution of the penalty term $P(\boldsymbol{\phi}_t)$, while the work in (Vlaski & Sayed, 2021) can directly study the evolution of the network centroid due to the problem being linear in the Euclidean case (the reason can be found in *[the response to Reviewer tsm9](https://openreview.net/forum?id=5tyvHfhRFZ&noteId=FQqPRWKwJt)*). For convergence, our results focus on the non-asymptotic evolution of the cost function, while the Euclidean counterpart (Chen & Sayed, 2012; Sayed et al., 2013) only studies the bound of MSD performance at the steady state. Also, their results can benefit from the ease of the linear space. We will add more detailed comparisons in the revised manuscript.
>
> ***Highlight the design of the Lyapunov function:***
>
> Thank you for pointing out this vague definition. The Lyapunov function in our case is defined by $\Delta_s'=\mathbb{E}[J(\boldsymbol{w}_t')-J(\boldsymbol{w}^*)]$. The design of the Lyapunov function is special in this context due to the manifold curvature. While in the Euclidean case one can use a Lyapunov function $\Delta_s=\mathbb{E}[J(\boldsymbol{w}_t)-J(\boldsymbol{w}^*)]$, in the Riemannian case when "telescoping" the decrease in $\Delta_s$ in the analysis, the curvature-related term $\zeta$ prevents the cancellation of intermediate terms and makes this approach unfeasible. Thus, we use a specially designed "curvature aware" Lyapunov function $\Delta_s'=\mathbb{E}[J(\boldsymbol{w}_t')-J(\boldsymbol{w}^*)]$ inspired by (Zhang & Sra, 2016), which is a function of the "streaming average" of the iterates denoted by $\boldsymbol{w}_t'$, as defined in (29). The idea consists of averaging the iterates carefully using the curvature-related parameter $\zeta$ to obtain the desired cancellation of terms in (84) when telescoping the decrease of $\Delta_s'$ in the convergence analysis (please see details in Appendix B.2 of the manuscript).
>
> We will revise the manuscript to clarify the definition of this Lyapunov function and the reason behind its choice.
>
> ***Report the experimental results on the step size choices:***
>
> Thank you for this suggestion. For our algorithm, the step sizes are indeed critical to the tradeoff between convergence speed and steady-state performance. We provide an illustrative experimental result as in https://ibb.co/Q7L00mzj, and will report more results in the revised manuscript.
>
> ***Fix the typo of zero denominators:***
>
> Thanks for pointing out this typo, the denominator should be $s-s_0+1$ as in Appendix B.2 of the manuscript.
>
> ***Fix mistakes in the proof of Lemma 5.1:***
>
> Thanks a lot for pointing out this mistake. The correct proof uses Jensen's inequality and include the missing condition that the adjacency matrix of the graph $C$ is left-stochastic, that is, $c_{\ell k}\geq 0, \sum_{\ell=1}^K  c_{\ell k} = 1$ for each agent $k$ in the assumption of "Regularization on graph".  This condition is fairly standard (Chen & Sayed, 2012; Sayed et al., 2013; Vlaski & Sayed, 2021) and can be assumed without loss of generality.
> The correct proof is produced below, and these modifications do not influence any other proofs of the manuscript, whose correctness has also been checked by us and other reviewers.
>
> _Proof_: From the definition of $\nabla P(\phi_t)$ and $P(\phi_t)$, we have
>
> $$
> \lVert\nabla P(\phi_t)\rVert^2 =  \sum_{k=1}^K\left\lVert- \sum_{\ell=1}^K  c_{\ell k} \exp_{\phi_{k,t}}^{-1}(\phi_{\ell,t})\right\rVert^2 \leq \sum_{k=1}^K\sum_{\ell=1}^K  c_{\ell k} \left\lVert\exp_{\phi_{k,t}}^{-1}(\phi_{\ell,t})\right\rVert^2 = 2P(\phi_t),
> $$
>
> where the inequality follows from Jensen's inequality and the assumption that $C$ is left-stochastic.

---

### Official Review · Reviewer_qfKq · 2025-03-24

**Overall Recommendation:** 4

**Summary:**

This paper presents a novel Riemannian generalization of a diffusion adaptation strategy for distributed optimization. The distributed optimization aims at finding an optimal solution with consensus among different agents. The proposed algorithm utilizes the Riemannian exponential map on manifolds and obtains a non-symptotic convergence under appropriate assumptions. In particular, a network agreement among the agents is guaranteed after sufficient iterations, which minimizes the (locally) convex risk function. The experimental results demonstrate the convergence when the Riemannian exponential map is replaced with an appropriate retraction map.

**Claims And Evidence:**

The claims made in this submission are accurately stated and supported by convincing evidence and discussions. Except for the following statement

``A work extending the diffusion strategy to manifolds was introduced in (Wang et al., 2024b), but the algorithm is inefficient due to inner-loop optimization ...'',
which is not supported by any experimental evidence.

>> REVIEW UPDATE: Authors have responded to this concern and included supporting experimental result.

**Essential References Not Discussed:**

This paper focuses on the diffusion adaptation strategy that utilizes the full gradient information and there is also a stochastic diffusion adaptation strategy proposed in

Nonconvex Federated Learning on Compact Smooth Submanifolds With Heterogeneous Data by Jiaojiao Zhang, Jiang Hu , Anthony Man-Cho So, Mikael Johansso, published in  Proceedings of the AAAI Conference on Artificial Intelligence.

It is worth mentioning the recent developments in Riemannian federated learning as a counterpart of decentralized algorithms for distributed optimization on manifolds. In particular, there are recent arXiv preprints:

Riemannian Federated Learning via Averaging Gradient Stream by Zhenwei Huang, Wen Huang, Pratik Jawanpuria, Bamdev Mishra.

Federated Learning on Riemannian Manifolds with Differential Privacy by Zhenwei Huang, Wen Huang, Pratik Jawanpuria, Bamdev Mishra.

**Experimental Designs Or Analyses:**

I checked the experimental designs and analyses. I find them organized well overall but I have some concerns that are not clearly addressed in:

``For computational simplicity, we replace the exponential maps in the updates (3) and (4) with approximate retractions.''

In addition to the concerns on the approximate retractions, I also believe that the potential of the proposed algorithm is not fully demonstrated in the analyses. While the Riemannian centralized method obtains the superior performances in terms of elapsed comput time, one of the most important advantages of decentralized optimization is the parallel computations over the agents. Accumulating the elapsed comput time blur the parallel computing advantages of the proposed algorithm.

Please refer to the ``Questions For Authors'' section that explain my concerns.

>> REVIEW UPDATE: Authors have clarified the concerns and confusions addressed above.

**Methods And Evaluation Criteria:**

The proposed methods and the evaluation criteria make sense for the problems of interest in this paper.

**Other Comments Or Suggestions:**

I do not have comments other than those that have been addressed in other sections.

**Other Strengths And Weaknesses:**

Overall, the algorithmic design and theoretical analysis of the novel Riemannian diffusion adaptation strategy proposed in this paper is significant. In particular, the guaranteed convergence result for the Riemannian diffusion adaptation strategy as well as the techniques used to prove it are essential for this rising topics of distributed optimization.

**Questions For Authors:**

The following questions/concerns stand out in my opinion.


1. The algorithmic design and theoretical analysis of the Riemannian diffusion adaptation strategy proposed in this paper are entirely based on the Riemannian exponential mapping. Why is this mapping replaced by the approximate retraction mapping in the numerical experiment? Does the theoretical analysis still apply to the approximated version of the diffusion strategy?
1. A much more important follow-on question: please be specific whether the approximate retraction mapping is employed in all algorithms that are being tested in the experiments. This is important because the experimental results are reported in terms of the elapsed compute time, which is significantly affected by the choice of update computation on manifolds.
1. What are the performance of the proposed method and other decentralized algorithms if the computations over agents are considered in parallel computing?


As stated above, the algorithmic design and theoretical analysis of the novel Riemannian diffusion adaptation strategy proposed in this paper is significant and my opinions on this submission leaning towards accept. While it is a common practice to relax the Riemannian exponential mapping with retraction mapping, this submission did not address the implications of the approximate retraction mapping used in the numerical experiment properly, which hurts the soundness of the numerical experiments as raised in the first two questions. Since the convergence results are actually obtained for the ``weaker'' retraction mapping, this paper is certainly recommended for ICML in my opinion if the first two questions are clearly addressed. The third question is more of a favourable question that exploits more potentials in the proposed algorithm. It would not damage my evaluation of this paper if it is not responded.

>> REVIEW UPDATE: Authors have clarified the concerns and confusions addressed above.

**Relation To Broader Scientific Literature:**

This paper proposes the first Riemannian diffusion adaptation strategy with guaranteed convergence for general manifolds. The proposed algorithm outperforms the existing Riemannian diffusion adaptation strategy [CGHS'21] specifically designed for PCA problem and generalized extrinsic consensus strategy from [NOP'10,LZZHZL'17] for GMM. This paper also claims bad performance of the Riemannian diffusion adaptation strategy introduced in [WBR'24b], but the claim is not supported by numerical evidence.

**Theoretical Claims:**

Yes, I checked and confirmed the correctness of the proofs. In particular, the proofs of Lemma 5.11, Theorem 5.12 and Theorem 5.15 are carefully examined.

---

> ### Author Rebuttal · Authors · 2025-03-30
>
> Thank you for reading our work (especially in checking the proofs) and offering constructive suggestions. In the following, we provide clarifications and answers to your comments and questions.
>
> ### **Replies to Weakness:**
>
> ***Support the claim that the method in (Wang et al., 2024b) is not efficient:***
>
> Thank you for this insightful comment. To support this claim, we compare the performance and runtime between the work in (Wang et al., 2024b) (denoted as "Inefficient Riemannian diffusion") and the proposed algorithm in the same setting as in Section 7.1 of the manuscript. We examine these two algorithms and produce the results as in https://ibb.co/nsDtry65. From these results, we can claim that while the performance of these two algorithms is nearly identical, the proposed algorithm achieves a significantly reduced runtime. We will add this supporting experiment to the revised manuscript.
>
> ***Discuss more Riemannian federated learning works:***
>
> Thank you for mentioning related work on Riemannian federated learning, which is an important rising topic and a counterpart of our research goal. We will add the suggested references to the revised manuscript.
>
> ### **Replies to Questions:**
>
> ***Why is the exponential mapping replaced by a retraction in the experiment? Does the analysis still apply?***
>
> We suggest replacing the exponential map with a retraction for computational reasons, as a retraction can be more efficient than the exponential map for certain manifolds (Boumal, 2023). While the exponential map is convenient for theoretical analysis, the retractions often lead to more practical and efficient computations. We will further clarify the motivation for this choice in the revised manuscript.
>
> Our theoretical analysis, like many works in Riemannian optimization, e.g., (Zhang & Sra, 2016), is based on the exponential map.
> A key result in (Bonnabel, 2013) states that $d(R_x(\mu\cdot v),\exp_x(\mu\cdot v))=O(\mu^2)$,  meaning that for small $\mu$, a retraction closely approximates the exponential map. The main approach to proving convergence with retractions involves showing that the iterates of the algorithm remain close to those of an equivalent version using the exponential map, which holds as $\mu \to 0$ (Bonnabel, 2013). This argument typically relies on diminishing step sizes, whereas our analysis is designed for constant step sizes, which are crucial for continuous adaptation and learning. Some works also employ the _pullback_ operator $f\circ R_x$, i.e., the composition of the cost function $f$ and a retraction, to establish convergence. However, these approaches requires assumptions that may be less natural, such as the convexity and smoothness of the pullback operator, see Chapter 4 of (Boumal, 2023). Thus, we believe that extending the proposed theoretical analysis based on a retraction is an exciting, though non-trivial research direction. We will discuss this limitation of the theoretical analysis in the revised manuscript.
>
> ***Whether the retraction is employed in all algorithms?***
>
> All the algorithms use the same retraction for a fair comparison.  The experimental results are reported in terms of _time_, representing the time index of receiving streaming data $\boldsymbol{x}_t$, which can also be regarded as the iteration index of the algorithm in the stochastic algorithms. To avoid possible ambiguity, we will replace "time" with "iteration" and update all the related figures in the revised manuscript.
>
> ***What is the performance in parallel computing?***
>
> The proposed method, like most decentralized algorithms, can benefit from parallelization. Given the response to the last question, all reported performance results (replacing "time" with "iteration") can be regarded as computations performed across agents in a parallel computing setting.
>
> This can also be seen when we analyze how the computational complexity scales with the number of agents $K$ in the network, keeping in mind that all operations (i.e., both the adaptation and combination steps) are fully parallelizable over the agents. For a detailed discussion on the computational complexity of the proposed algorithm, please refer to *[the response to Reviewer xKFw](https://openreview.net/forum?id=5tyvHfhRFZ&noteId=Lgg3fpLQRg)*.

---

### Decision · Program_Chairs · 2025-05-01

**Decision:**

Accept (poster)

**Comment:**

The paper has received positive reviews and all the reviewers are of the view of accepting the paper. In the final version, please take care of the comments and address the minor issues. Also include the explanations missing.